# Polycomb-mediated silencing of *miR-8* is required for maintenance of intestinal stemness in *Drosophila melanogaster*

Zoe Veneti [1,2] ✉, Virginia Fasoulaki[1,3], Nikolaos Kalavros [4,5], Ioannis S. Vlachos [4,5,6], Christos Delidakis [1,3] & Aristides G. Eliopoulos [7,8] ✉

Balancing maintenance of self-renewal and differentiation is a key property of adult stem cells. The epigenetic mechanisms controlling this balance remain largely unknown. Herein, we report that the Polycomb Repressive Complex 2 (PRC2) is required for maintenance of the intestinal stem cell (ISC) pool in the adult female *Drosophila melanogaster*. We show that loss of PRC2 activity in ISCs by RNAi-mediated knockdown or genetic ablation of the enzymatic subunit Enhancer of zeste, E(z), results in loss of stemness and precocious differentiation of enteroblasts to enterocytes. Mechanistically, we have identified the microRNA *miR-8* as a critical target of E(z)/PRC2-mediated tri-methylation of histone H3 at Lys27 (H3K27me3) and uncovered a dynamic relationship between E(z), *miR-8* and Notch signaling in controlling stemness *versus* differentiation of ISCs. Collectively, these findings uncover a hitherto unrecognized epigenetic layer in the regulation of stem cell specification that safeguards intestinal homeostasis.

Balancing maintenance of self-renewal and differentiation is a key stem cell property. The mechanisms controlling this balance and thus the regulation of downstream stem cell fate decisions remain fundamental questions in biology. Components of the epigenetic machinery emerge as important regulators of embryonic and adult stem cell fate specification. Among them, Polycomb group (PcG) proteins have attracted significant attention since their discovery as repressors of homeotic (*Hox*) gene expression that ensure the fidelity of developmental patterning in *Drosophila melanogaster*[1]. PcGs are now widely recognized for their role in controlling stem cell fate across organisms, ranging from plants to mammals, through chromatin remodeling and repression of gene expression[1,2].

PcG proteins combine to form Polycomb repressor complexes (PRCs). Thus, in *Drosophila*, the PcG proteins Enhancer of zeste (E(z)), Suppressor of zeste 12 (Su(z)12), and Extra sex combs (Esc) associate to generate the Polycomb repressive complex 2 (PRC2) which mediates trimethylation of lysine 27 of histone H3 (H3K27me3) through its enzymatic subunit E(z)[3–5]. In mammals, two complexes termed PRC2.1 and PRC2.2 are generated through the interaction of the core PRC2 components EZH2, SUZ12 and EED, the homologs of E(z), Su(z)12 and Esc respectively, with additional accessory proteins[6]. PRC2.1 possesses higher affinity to chromatin, leading to increased H3K27me3 deposition and silencing of PcG target genes, compared to PRC2.2[7].

The physiological role of PRC2 in developmental processes in humans is underscored by loss-of-function mutations in *EZH2*, *EED*, and *SUZ12* associated with Weaver-Smith (OMIM #277590), Cohen-Gibson (OMIM #617561) and Imagawa-Matsumoto (OMIM #618786) syndromes, respectively. These genetic disorders share several clinical characteristics, including variable prenatal and postnatal overgrowth and musculoskeletal abnormalities, dysmorphic features and impaired

[1]Institute of Molecular Biology and Biotechnology, Foundation of Research & Technology Hellas, Heraklion, Greece. [2]Medical School, University of Crete, Heraklion, Greece. [3]Department of Biology, University of Crete, Heraklion, Greece. [4]Spatial Technologies Unit, Harvard Medical School Initiative for RNA Medicine, Department of Pathology, Beth Israel Deaconess Medical Center, Boston, MA, USA. [5]Broad Institute of MIT and Harvard, Cambridge, MA, USA. [6]Harvard Medical School, Boston, MA, USA. [7]Laboratory of Biology, School of Medicine, National and Kapodistrian University of Athens, Athens, Greece. [8]Center of Basic Research, Biomedical Research Foundation of the Academy of Athens, Athens, Greece. ✉e-mail: veneti@imbb.forth.gr; eliopag@med.uoa.gr

intellectual development[7]. In vitro studies have demonstrated that these *EZH2*, *EED*, and *SUZ12* germline mutations reduce H3K27 methylation, suggesting a causative link between loss of PRC2 function, chromatin remodeling and developmental abnormalities in humans[8,9]. In somatic cells, deregulation of EZH2 has been linked to malignancy[10]. Depending on the tumor type, both oncogenic and tumor-suppressor functions have been ascribed to EZH2, reflecting the pleiotropic effects of PRC2 on chromatin structure and gene expression[11].

The intestinal epithelium is the most vigorously self-renewing tissue in numerous adult organisms. The *Drosophila* intestine is maintained and regenerated by resident intestinal stem cells (ISCs) which are the predominant mitotic cells in this tissue and give rise to the absorptive enterocyte (EC) and the secretory enteroendocrine (EE) cell lineages. Lineage commitment to ECs is mediated through transit post-mitotic progenitor cells called enteroblasts (EB). This process requires signaling between Delta (Dl) expressed in the daughter cell that remains a stem cell and Notch expressed in the daughter cell that becomes the committed EB[12–14]. In the homeostatic intestine, a pool of EBs is thus generated which remains incompletely differentiated until EC renewal is needed[15]. The transition of EB to EC entails two or three cycles of genome endoreplication accompanied by a progressive increase in cell size and the development of a brush border on the apical side of ECs[16]. In contrast, the EE progenitors (pre-EEs) can be generated directly by ISCs upon transient activation of the Scute (*Sc*) transcription factor, which typically divide once more before their terminal differentiation to produce a pair of EE cells[12]. Whereas Notch promotes the EB fate when the ISC divides to generate new ECs, it inhibits *sc* expression and EE commitment in the ISC to pre-EE divisions[17].

Even though PRC2 regulates the expression of several developmental genes involved in germline stem/progenitor cell physiology in male *Drosophila*[18], the in vivo actions of E(z)/PRC2 in ISC fate decisions and lineage progression have not yet been elucidated other than a role in EE fate specification[19]. In mice, Koppens et al.[20] and Chiacchiera et al.[21] explored conditional deletion of EED in the intestine to disrupt PRC2. They reported that loss of PRC2 function in the entire intestinal epithelial population favors the accumulation of secretory lineage cells and identified Atonal homolog-1 (Atoh1), a transcription factor required for secretory cell commitment downstream of Notch signaling[22], as a relevant PRC2 target. Albeit informative, these mouse studies do not address the function of PRC2 components in specific intestinal cell populations, including ISCs.

Herein, we have combined genetics with lineage-tracing methods in *Drosophila* to decipher the role of E(z)/PRC2 in ISCs under normal and pathological conditions. We report that disruption of E(z) function specifically in ISCs leads to loss of stemness and a differentiation bias toward the EC lineage. Moreover, we identify *miR-8* (*miR-200* in humans) as a PRC2 target and reveal a dynamic relationship between E(z), *miR-8* and Notch signaling in controlling stemness vs. differentiation of ISCs. Thus, our findings uncover a hitherto unrecognized epigenetic layer in the regulation of intestinal stem cell specification.

## Results

### E(z) is required for the maintenance of ISC pool and regenerative capacity in the Drosophila midgut

To define the role of E(z) in the adult Drosophila midgut we knocked down E(z) in intestinal progenitor cells (ISCs and EBs) by activating RNAi expressed under the control of the conditional, temperature-sensitive esg-Gal4, upstream activating sequence (UAS)-GFP, tub-GAL80ts system[13]. The escargot (esg) Gal4 driver is active in both ISC and EB but not in differentiated midgut cells (EE and EC). Thus, once shifted to the non-permissive temperature, RNAi in esg-positive cells is induced, and ISCs/EBs are simultaneously marked by esg-Gal4 driven GFP. The efficacy of the in vivo knock-down of *E(z)* was confirmed by RT-qPCR using RNA isolated from FACS-sorted GFP-positive (GFP[+]) intestinal progenitors (Supplementary Fig. 1A, B).

Following *E(z)* depletion for 4 days, we witnessed a marked decrease in the number of GFP[+] cells (Fig. 1A, C, E) and reduced numbers of cells stained positive for the phosphorylated form of Histone-3 (pH3[+]), a commonly used mitotic marker, suggesting that ISC proliferation is affected (Fig. 1F).

We hypothesized that knock-down of *E(z)* may similarly affect ISC proliferation upon intestinal damage and thus aggravate intestinal pathologies. To this end, E(z)-RNAi and control flies were fed with Dextran Sulfate Sodium (DSS), a chemical that causes disruption of intestinal basement membrane organization leading to compensatory stem cell proliferation necessary for tissue regeneration[23]. In mice, oral administration of DSS instigates a cascade of events associated with the physical disruption of the mucosal barrier of the large intestine that leads to ulcerative colitis-like pathological manifestations[24]. In line with a previous report[23], administration of DSS in control flies increased both esg>GFP[+] and pH3[+] cells compared to sucrose-feeding, reflecting compensatory intestinal tissue regeneration. However, these effects were ameliorated upon knock-down of *E(z)* (Fig. 1A–F). In addition, E(z)-RNAi flies fed with DSS exhibited lower survival rates than their respective controls (Fig. 1H), suggesting that E(z) impacts intestinal homeostasis and regeneration after damage. Similar phenotypes were obtained in two additional E(z)-RNAi lines and in flies with ISC/EB-specific knock-down of *Su(z)12* or *esc* (Supplementary Fig. 1C). The isogenic strain w[1118] was crossed to the esg-Gal4 driver and used as control for all RNAi experiments. Two unrelated UAS transgenes (lacZ or w-RNAi) were compared to w[1118] to eliminate the possibility that the E(z)-RNAi phenotype was an artifact due to competition for the Gal4; no differences were observed with respect to the percentage of GFP+ progenitors, pH3[+] mitotic cells and survival upon sucrose or DSS-fed conditions (Supplementary Fig. 2).

To determine whether E(z) acts predominantly in ISCs, EBs or both progenitors, we knocked down E(z) by an EB-specific Gal4 (Su(H) GBE-Gal4, UAS-GFP; tub-Gal80ts[25]) or an ISC-specific Gal4 (esg-Gal4, UAS-GFP; Su(H)GBE-Gal80, tub-Gal80ts[26]). Flies were fed and processed as in Fig. 1. The results showed that the knock-down of E(z) in ISCs, but not EBs, reproduces the effect of E(z) depletion in intestinal progenitor cell number and mitoses (see Supplementary Fig. 3 vs. Fig. 1).

To further explore the fate of ISCs upon E(z) knock-down, we dissected Drosophila midguts and performed immunostaining for Delta (Dl), a specific marker for ISCs, and Prospero (Pros) which is expressed specifically in EE cells. Delta positive (Dl[+]) and Prospero positive (Pros[+]) cells were scored and normalized to DAPI[+] cells. At 7 days of E(z)-RNAi induction, we observed a significant decrease in the number of Dl[+] (punctate staining in the cytoplasm of GFP[+] cells) but not of Pros[+] cells (solid nuclear staining in GFP-negative cells) (Fig. 2). However, the number of Pros[+] EE cells were markedly reduced following 15 days of E(z)-RNAi induction (Supplementary Fig. 4A, B)[19], likely reflecting a secondary outcome of ISC depletion. Morphologically, the midguts of these E(z)-knocked down aged flies appeared leaky, exhibiting large areas of epithelial layer disruption and extensive penetration of bacteria that was evident by DAPI staining. Few Dl[+] cells were detected that contained traces of GFP (Supplementary Fig. 4C) and were often larger than control ISCs with a nucleus diameter of more than 7 μm, resembling polyploid ECs (Supplementary Fig. 4C). By staining exclusively for Dl and DAPI, we confirmed fewer Dl[+] cells after 10 days of E(z)-RNAi compared to w-RNAi induction (Supplementary Fig. 4D, E).

Collectively, these results suggest that E(z)/PRC2 is required for the maintenance of a functional ISC pool that is critical for regenerative capacity in the Drosophila midgut.

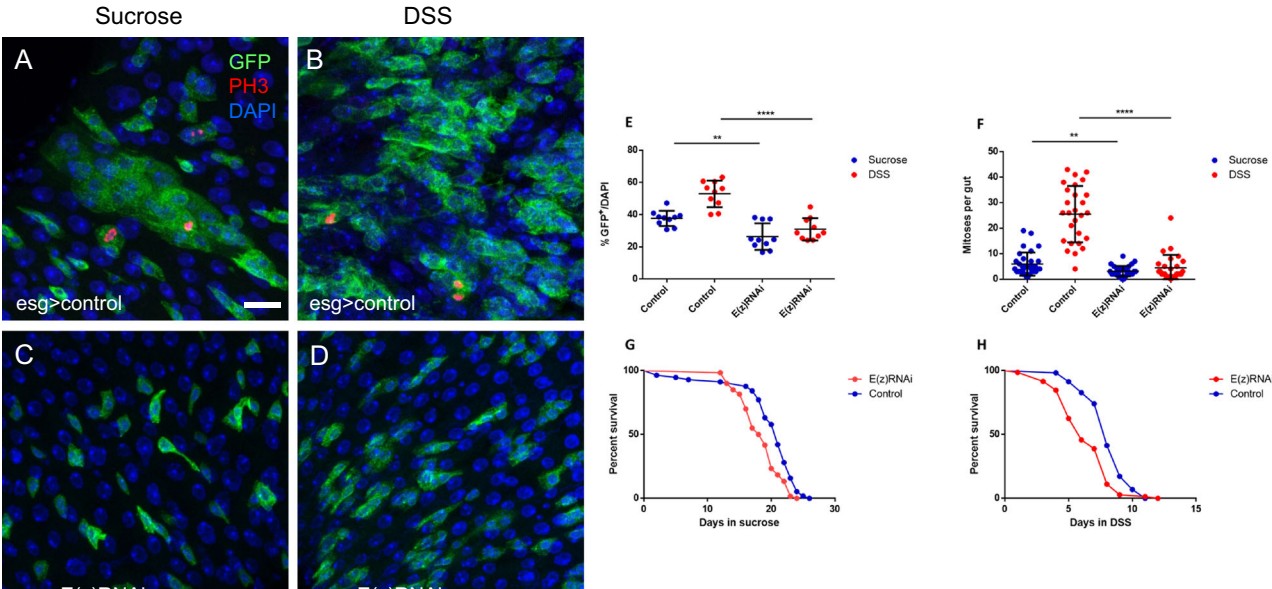

**Fig. 1 | E(z) is required for ISC proliferation and midgut regeneration in Drosophila.** Immunofluorescence phospho-Histone 3 (pH3) staining of posterior midguts from 7 day old w[1118] control adult female flies expressing esg-Gal4, tub-Gal80ts, UAS-GFP (esg[ts] > GFP) fed with 5% sucrose solution (**A**) or 3% DSS (**B**) for 2 days prior to dissection; ISCs/EBs (GFP, green), mitotic cells (pH3, red), nuclei (DAPI, blue). Note the increase in the percentage of GFP[+] cells after DSS treatment in (**B**) vs. (**A**). Young (3 days old) mated females were shifted to 29 °C for 2 days to express E(z)-RNAi and then fed with sucrose (**C**) or DSS (**D**) for two additional days at 29 °C before being processed as described in (**A**, **B**). Note the absence of the DSS-induced increase in the percentage of GFP[+] cells in (**D**) vs. (**B**). Scale bar; 25 μm, applicable to (**A–D**) images. **E** Quantification of the GFP[+] cells as a percentage of total DAPI-stained cells. Data are represented as mean ± SD. Two-tailed, unpaired *t*-test: *p* = 0.0014 for sucrose and *p* < 0.0001 for DSS fed flies.

Zoom-in images from ten posterior midguts were randomly selected for counting. **F** Quantification of pH3[+] positive cells representing mitotic ISCs in control and E(z)-RNAi midguts. Data represented as mean ± SD. A decreased number of pH3[+] cells was observed upon induction of E(z)-RNAi after both sucrose (Two-tailed, unpaired *t*-test, *p* = 0.0029) and DSS (*p* < 0.0001) treatment. Cells were counted from whole guts, *n* = 32 for control flies fed with sucrose, *n* = 30 for E(z)-RNAi flies fed with sucrose, *n* = 27 for control flies fed with DSS and *n* = 27 for E(z)-RNAi flies fed with DSS, pooled from three independent experiments. **G**, **H** Survival curves for control and E(z)-RNAi flies fed with sucrose (*p* = 0.0004) or DSS (*p* < 0.0001; log-rank (Mantel−Cox) test). The number of flies (*n*) assessed is *n* = 57 for control flies fed with sucrose, *n* = 60 for E(z)-RNAi flies fed with sucrose, *n* = 58 for control flies fed with DSS and *n* = 72 for E(z)-RNAi flies fed with DSS, pooled from three independent experiments.

## Loss of E(z) in ISCs induces precocious differentiation into ECs

On the basis of the aforementioned observations we examined whether the effect of E(z) depletion in the ISC pool might reflect direct inhibition of proliferation, induction of cell death or precocious differentiation.

First, we performed terminal deoxynucleotidyl transferase dUTP nick end labeling (TUNEL) staining to quantify cell death. We did not observe apoptotic cells in the progenitor population irrespective of genotype and treatment (Supplementary Fig. 5), suggesting that the effect of E(z) depletion on ISC number does not involve cell death induction.

To test for possible effects on differentiation, the mosaic analysis with a repressible cell marker (MARCM) methodology[27] was used. By this technique, ISC homozygous clones for the wild type or null allele of *E(z)*[731] were randomly generated by a heat shock-induced FRT-recombination event that simultaneously led to induction of GFP expression. The number of GFP[+] cells in wild type and E(z)[731] clones was monitored at 7, 12 and 19 days after induction of recombination by fluorescence microscopy (Fig. 3 and Supplementary Fig. 6). At 7 days, the wild type and mutant clones were about the same size. At 12 and 19 days after induction of recombination, the wild type clones contained more than 10 GFP[+] cells, including both large and small-size intestinal cells (Fig. 3A, A', C and Supplementary Fig. 6A, B). In contrast, MARCM clones that were generated from ISCs bearing the null allele of *E(z)*[731] contained a limited number of GFP[+] cells (a maximum of 8 cells per clone) that had mostly a large diameter (Fig. 3B, B', C and Supplementary Fig. 6). MARCM analysis of mutated *Su(z)12*-carrying clones produced similar results (Supplementary Fig. 7).

Immunofluorescence staining for the markers Delta (Dl) which detects ISCs, Prospero (Pros) expressed in EE, and Pdm1 expressed in differentiated ECs, revealed that the *E(z)*[731] clones contained mostly Dl-negative, Pros-negative and Pdm1-positive cells (Fig. 3D–F and Supplementary Figs. 6C and 8C). Further experiments showed that *E(z)*[731] mutant clones did not contain any Esg-positive cells (Supplementary Fig. 8B, C) indicating loss of progenitor cells. Taken together, the aforementioned data suggest that loss of E(z)/PRC2 function in ISCs leads to progressive reduction in stem cell numbers due to a differentiation shift toward the EC lineage.

## The knock-down of *E(z)* in intestinal progenitor cells promotes the transition of enteroblasts to enterocytes

To pinpoint the differentiation defect caused by *E(z)* knock-down in ISCs, we used the Gal4 technique for real-time and clonal expression (G-TRACE), a lineage-tracing method developed in *Drosophila*[28]. To initiate G-TRACE in midgut, esg-driven Gal4 expressed in ISCs and EBs was used to induce the expression of RFP and FLP recombinase, which removes an FRT-flanked transcriptional termination cassette inserted between a Ubiquitin-p63E (*Ubi*) promoter fragment and the open reading frame of nuclear EGFP (*nEGFP*) gene (*UAS-Flp, UAS-RFP, Ubi-FRT-STOP-FRT-nEGFP*). As a result, progeny cells are permanently marked with nuclear EGFP whereas progenitor cells (ISCs and EBs) express RFP because of the conditional, temperature-sensitive driver *esgGal4 Gal80[ts]*.

In control flies fed with sucrose or DSS for 36 h after G-TRACE activation caused by shifting temperature to 29 °C to dis-inhibit Gal4, newly produced ECs were scarcely detected (Fig. 4A, B, D). This is in

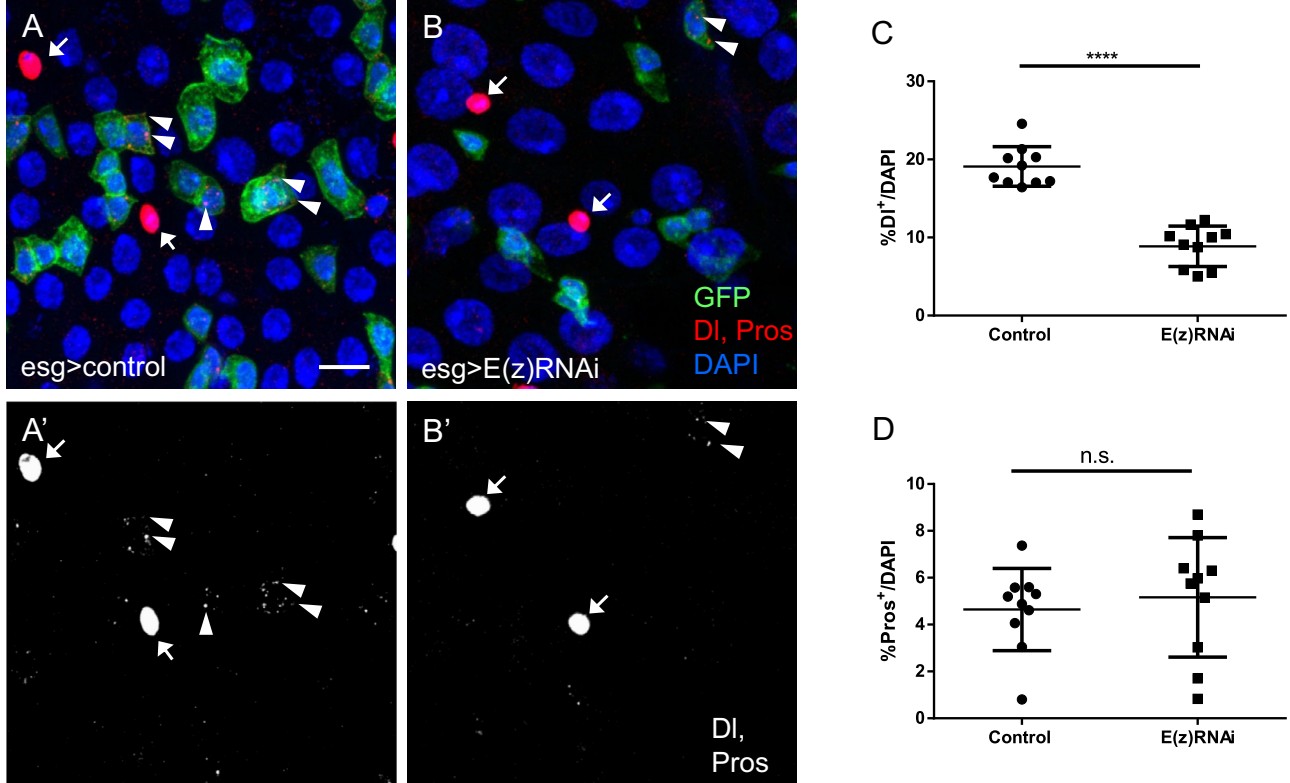

**Fig. 2 | E(z) depletion leads to loss of ISCs.** Flies with esg^ts > GFP were crossed to either UAS-E(z)-RNAi to knockdown E(z) in intestinal progenitor cells or to w[1118] to generate the respective control genotype (control). **A, B** Control and E(z)-RNAi posterior midguts were immunostained at 7 days post induction for the ISC marker Delta (Dl, cytoplasmic puncta in GFP[+] cells, arrowheads) and the EE marker Prospero (Pros, solid nuclear staining in GFP-negative cells, arrows). Scale bar; 20 μm, applies to all images. **C, D** Quantification of Dl[+] and Pros[+] cells as percentages of total DAPI-stained cells (Two-tailed unpaired $t$-test: $p < 0.0001$ for Dl[+] cells). Randomly selected posterior midgut zoom-in images from each of $n = 10$ flies were analyzed per marker and genotype. Data are represented as mean ± SD.

agreement with a previous report showing that DSS induces accumulation of "stalled" EBs[23]. The EBs of the DSS-fed control flies resumed differentiation following 4 days recovery in normal feeding conditions (Fig. 4C, D), as previously reported[16].

In contrast, G-TRACE activation in E(z)-RNAi flies displayed a marked increase in newly generated ECs, which amounted to ~20% of the DAPI[+] cells in the R4 region of posterior midgut in either feeding condition (Fig. 4A, B, D). This was further increased following recovery from DSS (Fig. 4C, D). Overall, the aforementioned data demonstrate that E(z) depletion accelerates differentiation of EBs to ECs under normal and pathological conditions.

### E(z) depletion reduces chromatin-bound H3K27me3 and leads to the up-regulation of *miR-8* expression

To identify direct targets of E(z)/PRC2 in ISCs/EBs that may underlie the aforementioned phenotypes, we conducted genome-wide mapping of chromatin-bound H3K27me3 using chromatin immunoprecipitation sequencing (ChIP-seq). First, we assessed H3K27me3 expression in the normal intestine using immunofluorescence microscopy. As shown in Supplementary Fig. 9A, C, H3K27me3 is readily detected in ISCs/EBs whereas expression is variable in ECs. The RNAi-mediated knock-down of E(z) and the loss-of-function E(z)[731] mutant reduced total H3K27me3 in intestinal cells (Fig. 5A and Supplementary Fig. 9C). These analyses also showed elevated levels of nuclear H3K27me3 in intestinal progenitor cells of DSS-fed control flies which, however, were diminished when E(z) was knocked-down (Fig. 5A). We also observed elevated levels of H3K27me3-specific fluorescence in ISCs/EBs of aged compared to young flies (Supplementary Fig. 9A, B).

Next, a pool of midguts dissected from at least 100 control or E(z)-RNAi flies were FACS-sorted to isolate GFP-expressing ISCs/EBs that were processed for H3K27me3 ChIP-Seq. Differential enrichment analysis identified 179 chromatin regions with significant enrichment of H3K27me3 marks in the control compared to the E(z)-RNAi line (Fig. 5B and Supplementary Data 1). The *l(2)gl*, *ph-p*, *Psc* and *miR-8* loci displayed the largest reductions in H3K27me3 marks upon E(z) knockdown (Supplementary Data 2), suggesting that they are directly bound and repressed by E(z)/PRC2. By RT-qPCR we confirmed this result and showed that GFP[+] ISCs/EBs from E(z)-RNAi flies express higher RNA levels of *l(2)gl*, *ph-p* and *Psc* (Fig. 5C) as well as *miR-8* (Fig. 5D) compared to progenitor cells with intact E(z). For *miR-8* we tested both strands of the *miR-8* precursor (3p and 5p, Fig. 5D), because of previous reports suggesting that both the mature *miR-8-3p* and the passenger *miR-8-5p* have functional roles in Drosophila[29,30].

### *miR-8* is a functional target of E(z)/PRC2

*miR-8* has been implicated in the control of EB differentiation to ECs in the *Drosophila* intestine[15]. In particular, it has been shown that *miR-8* is upregulated in late EBs, shortly before terminal differentiation, and triggers terminal differentiation by antagonizing the snail-type transcription factor Escargot (Esg), which promotes the ISC/progenitor fates[15]. Thus, overexpression of *miR-8* in midgut progenitor cells induces their premature differentiation whereas *miR-8* depletion by a *miR-8* sponge causes their retention in an undifferentiated state[15]. In line with this report, we observed decreased ISC mitotic rates after *miR-8* overexpression, and over-accumulation of undifferentiated cells after depletion of *miR-8*, especially upon DSS treatment (Supplementary Fig. 10).

To define the in vivo effects of E(z) knockdown on *miR-8* expression, we used a *miR-8* EGFP sensor transgene[31] to monitor *miR-8* expression in control and E(z)-depleted midguts. The sensor

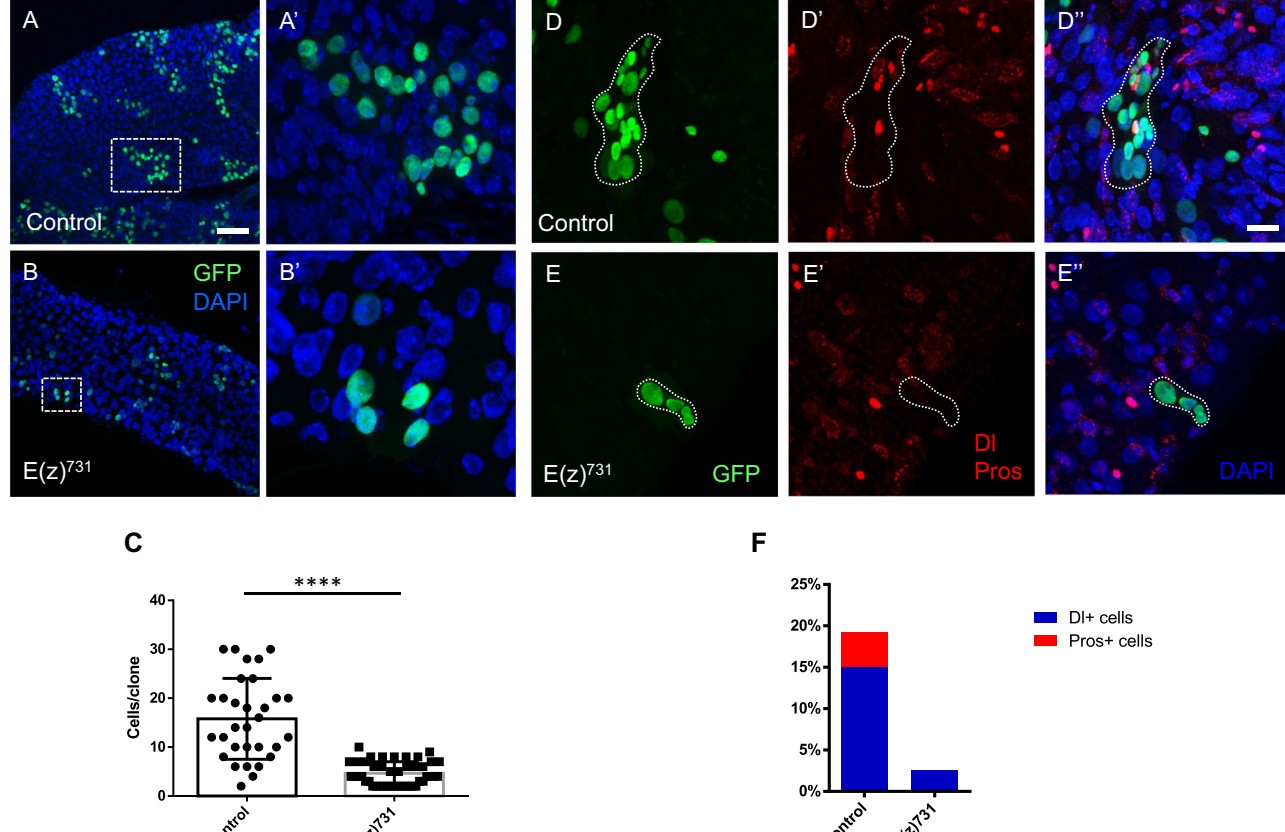

**Fig. 3 | Loss of E(z) function induces precocious differentiation into ECs. A–B'**
Midguts from adult flies containing nucleus-localized GFP-labeled control MARCM clones (**A–A'**) or clones of the catalytically inactive allele *E(z)731* (**B–B'**), stained for DAPI (blue). Guts were dissected from adult flies 10 days after clone induction. Scale bar; 50 μm. A higher magnification for each of the squared areas in (**A**) and (**B**) is depicted in (**A'**) and (**B'**) respectively. **C** Quantification of the cell numbers of the control or inactive *E(z)731* allele-carrying clones of the indicated genotypes. Results shown are the means ± SD of *n* = 31 for control and *n* = 44 for *E(z)731* clones (*p* < 0.0001, two-tailed unpaired *t*-test). **D–E"** Immunofluorescence staining of control (**D–D"**) and E(z)731 (**E–E"**) MARCM clones for Delta (Dl) and Prospero (Pros). Control clones have at least one Dl+ cell per clone and roughly 25% of the clones contain one Pros+ cell, whereas *E(z)731* clones rarely contain a Dl+ cell and never a Pros+ cell. Scale bar; 25 μm. **F** Quantification of Dl and Pros-positive cells from *n* = 237 cells/15 clones for control and *n* = 87 cells/20 E(z)731 clones.

transgene contains EGFP with two binding sites for *miR-8* in its 3'UTR, driven by the α-tubulin promoter. Thus, an increase in endogenous *miR-8* would lead to decreased expression of the EGFP sensor[31]. Following knock-down of E(z) we observed a reduction in EGFP-positive cells compared to control flies (Fig. 6A) suggesting upregulation of *miR-8*. To determine the nature of the *miR-8* expressing cells affected by E(z) depletion, we immunostained midguts of *miR-8* EGFP sensor transgenes with Dl and Pros antibodies and observed lower EGFP expression in Dl+ ISCs of E(z)-RNAi flies (Fig. 6B).

By immunofluorescence, we also evaluated H3K27me3 expression in the midgut of young *miR-8* EGFP sensor flies and found high EGFP associated with high H3K27me3 fluorescence in ISCs and EBs but absence of EGFP in ECs irrespective of H3K27me3 levels (Supplementary Fig. 11A). Conversely, EE cells displayed high levels of EGFP, but variable levels of H3K27me3. Because aging is associated with elevated H3K27me3 expression levels in ISCs/EBs (Supplementary Fig. 9A, B), we analyzed H3K27me3 levels in young (2–3 days old) vs. old (21 days old) *miR-8* EGFP sensor flies and found higher EGFP in intestinal progenitor cells in the latter (Supplementary Fig. 11B).

As the phenotype of *miR-8* overexpression is similar to that of *E(z)* knock-down in promoting enteroblast to enterocyte transition (Fig. 4 and Supplementary Fig. 10), and because E(z) depletion leads to elevated *miR-8* levels in the ISC/EB pool (Figs. 5D and 6B), we tested the hypothesis that E(z) mediates its effects through *miR-8*. We examined if the stemness defect caused by loss of E(z) in ISCs/EBs could be rescued by loss of *miR-8*. To address this question, we combined the

*miR-8* sponge with the MARCM strategy described in Fig. 3 and found that the failure of the E(z)731 mutant clones to expand was partially restored by simultaneous downregulation of *miR-8* (Fig. 6C, D). We conclude that *miR-8* is a functional target of E(z)/PRC2 in intestinal progenitor cells.

We further asked if the reverse association between *E(z)* and *miR-8* expression observed in the *Drosophila* intestine is conserved in humans. By mining the Human Protein Atlas database for single cell RNAseq data of normal human rectum and small intestine, we observed higher levels of EZH2 in the proliferative, Ki67-positive stem cell compartment (Supplementary Fig. 12A, B). In contrast, enterocytes expressed significantly lower levels of both Ki67 and EZH2. We also mined miRNA expression data in cells comprising the human small intestine[32] and found reduced levels of *miR-200a-5p*, a human homolog of *miR-8*, in ISCs compared to enterocytes (Supplementary Fig. 12C).

Additionally, differential expression analysis of The Cancer Genome Atlas Colorectal Adenocarcinoma (TCGA-COAD) samples[33] with the Genome Tissue Expression (GTEx) dataset revealed higher levels of EZH2 and RBBP7 in comparison to healthy tissue from GTEx[34] and adjacent normal samples from TCGA (Supplementary Fig. 13A). In contrast, the levels of *miR-200a* and *miR-200b* are reduced in colorectal tumors compared to normal tissue[35] (Supplementary Fig. 13B). Therefore, similar to *Drosophila* midgut, a reverse association between *EZH2* and *miR-200* expression is observed in both normal and malignant human intestine.

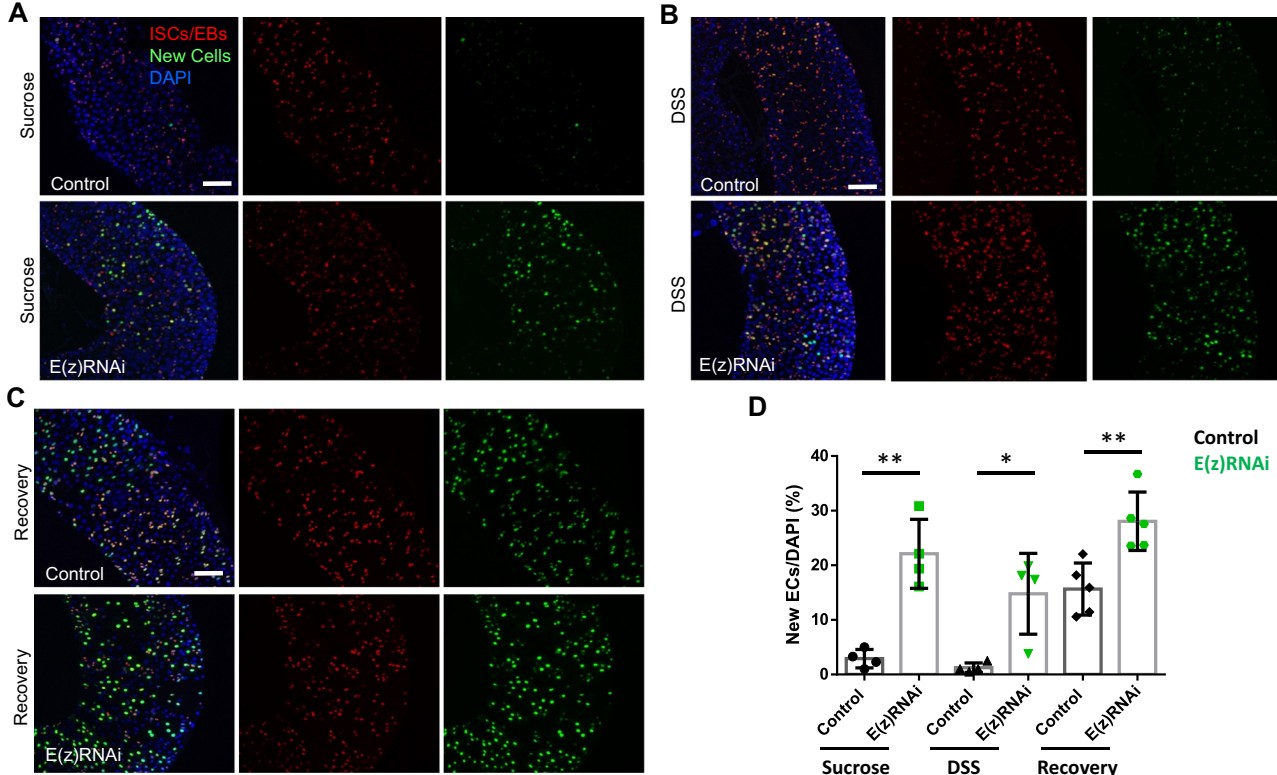

**Fig. 4 | Knock-down of *E(z)* promotes EB to EC transition.** The G-TRACE lineage tracing reporter is driven by esg-Gal4-tubGal80ts to label progenitor cells with UAS-RFP and newly produced progeny cells with nuclear EGFP, via the expression of UAS-FLP and Ubi-p63EFRT-stop-FRT-EGFP. Flies carrying these transgenes were crossed either to w[1118] (control) or to UAS-E(z)-RNAi. Collected progeny was aged for 4–7 days after eclosion and expression of transgenes was induced by 29 °C temperature shift. Newly differentiated ECs are identified as large cells expressing nuclear EGFP and low/no RFP. In control flies fed with sucrose (**A**) or DSS (**B**) for 36 h at 29 °C, newly produced ECs are scarcely detected in the R4 region of posterior midgut, whereas flies expressing E(z)-RNAi displayed a marked appearance of new ECs. Scale bars; 50 µm. **C** Flies were fed with DSS for 36 h to induce ISC proliferation and then left to recover under normal feeding conditions for 4 days. Control flies resume production of new ECs after 4 days of recovery from DSS to normal feeding conditions which is further increased upon E(z) knock-down. Scale bar; 50 µm. **D** Quantification of newly produced ECs per genotype and treatment. The number of large size cells with nuclear EGFP were counted from R4 posterior midguts and normalized to 100 cells stained with DAPI ($n = 4$ for control and E(z)-RNAi flies fed with sucrose or DSS, $n = 5$ for flies left to recover after DSS treatment). Values are presented as mean ± S.D. Two-tailed, unpaired *t*-test was used for statistical analysis: **$p = 0.0011$ (sucrose), *$p = 0.01$ (DSS), **$p = 0.0047$ (recovery after DSS).

## Notch is required for loss of stemness caused by *E(z)* depletion

Notch signaling is a key regulator of differentiation in the Drosophila ISC lineage[14,36]. Ectopic activation of Notch in ISCs/EBs results in precocious differentiation with a bias toward the EC fate, a phenotype similar to loss of E(z), whereas loss of Notch function leads to the accumulation of EE cells and ISC-like cells that express high levels of Delta, have high mitotic index and fail to differentiate into ECs[36]. Furthermore, Esg is required to maintain stem cell characteristics in these cells, including proliferative capacity, and to prevent differentiation toward the EE fate[37,38].

To detect putative epistatic relationships between Notch and E(z), we created wild type or homozygous *E(z)*[731] null allele MARCM clones that also expressed Notch RNAi. We confirmed[36] that the knock-down of Notch in midgut progenitors bearing wild type *E(z)* alleles led to dramatic expansion of MARCM clones that contained ISC-like cells (Fig. 7A) expressing high levels of Dl and Pros+ EE cells (Fig. 7B–D). Interestingly, whereas MARCM clones bearing *E(z)*[731] null alleles lacked ISCs and contained only a small number of differentiated ECs (Fig. 3), the simultaneous depletion of Notch was found to restore the number of cells per clone (Fig. 7E, I, J). The loss of ISCs was also rescued: Notch-RNAi in mutant *E(z)*[731] MARCM clones resulted in accumulation of intestinal stem-like cells expressing Dl, and a limited number of differentiated ECs or EE cells (Fig. 7F–H). The functional link between E(z) and Notch is further corroborated by the increased lifespan of double esg-GAL4 driven Notch-RNAi and E(z)-RNAi flies compared to Notch-

RNAi alone (Fig. 7K). Collectively, these observations indicate that the precocious differentiation of *E(z)* mutant ISCs requires the operation of Notch signaling.

To test whether the aforementioned effect of Notch-RNAi upon *E(z)* loss of function is mediated through modulation of *miR-8* levels, we employed the *miR-8* EGFP sensor transgene in single and double *E(z)* and *Notch* knockdowns. Compared to control flies (see Fig. 6A, B), Notch-RNAi was associated with increased expression of sensor EGFP, indicative of suppression of *miR-8* expression. Whereas depletion of *E(z)* alone led to higher *miR-8* expression levels compared to control flies (reduced sensor EGFP; Fig. 6A, B), the dual knockdown of *E(z)* and *Notch* was found to increase the number of EGFP+ cells with small nuclei (Supplementary Fig. 14), indicative of a decrease in *miR-8* expression. Therefore, *miR-8* cannot be expressed in the absence of Notch despite having lost the H3K27me3 mark. This effect is not complete, as some cells in the double knockdown escape and become polyploid, sensor-negative (i.e., *miR-8*-expressing) cells, presumably being ECs.

## Discussion

Polycomb group proteins maintain transcriptional repression both during embryonic development and in adult organs by restricting the activation of key factors involved in stemness and differentiation[39]. Evidence from mouse studies supports this concept by showing that generalized loss of PRC2 in the intestinal epithelium leads to the

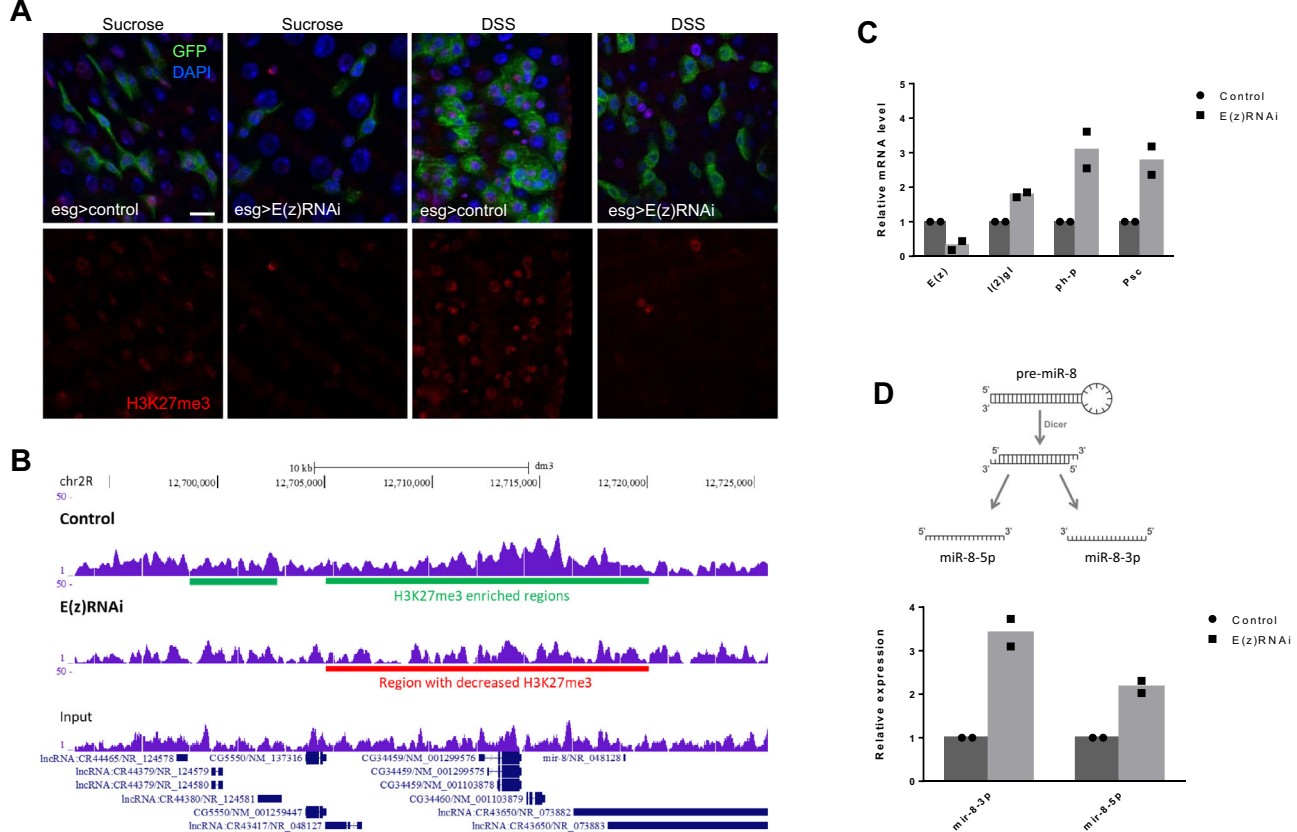

**Fig. 5 | E(z) depletion reduces chromatin-bound H3K27me3 and leads to the up-regulation of *miR-8* expression. A** Immunofluorescence staining for H3K27me3 (red) in control and E(z)-RNAi midguts after 7 days of RNAi induction and 2 days of feeding with sucrose or DSS. Scale bar; 25 µm. **B** Genome browser snapshot of a region in chromosome 2R containing the *miR-8* locus, where H3K27me3 was found reduced in E(z)-RNAi vs. control sorted GFP-expressing progenitor cells. **C** Quantification of *l(2)gl, ph-p* and *Psc* mRNA levels in ISCs/EBs depleted of E(z) relative to E(z)-proficient cells. Total RNA was extracted from two biological replicates of FACS-sorted GFP⁺ progenitor cells from 100 control and 100 E(z)-RNAi fly midguts for each replicate and expression levels were assessed by reverse transcription qPCR normalized to *rp49*, as indicated. **D** *miR-8-3p* and *miR-8-5p* levels increase upon E(z)-RNAi induction. Expression was normalized to *2S* rRNA. The origin of 3p and 5p is shown in the upper panel of (D).

accumulation of differentiated secretory cells due to de-repression of the secretory lineage master regulator *Atoh1*[20,21]. Although these findings indicate important contributions of PRC2 to cell fate decisions, they do not ascribe cell-specific roles of PRC2 in the regulation of intestinal homeostasis. Indeed, loss of PRC2 in the entire intestinal epithelium was reported to preserve ISC function in one study[21] but to impair Wnt signaling and ISC viability in another[20]. This ambiguity may reflect the impact of complex, still unresolved interactions among different cell lineages in the intestine arising from the generalized loss of PRC2 function. Along these lines, our in silico analysis of normal human rectum and small intestine single cell RNAseq data suggests that although EZH2 expression is highest in proliferative (Ki67-positive) stem cells and lowest in distal enterocytes, intermediate levels of expression are detected in other cell types (Supplementary Fig. 12A, B).

Herein, we have employed genetic and lineage-tracing methods in *Drosophila* to decipher the physiological role of E(z)/PRC2 specifically in ISC/EB progenitor cells. We have found that loss of PRC2 activity in ISCs results in loss of stemness associated with precocious differentiation but not increased ISC death (Fig. 1 and Supplementary Fig. 3). The consequences of PRC2 downregulation are particularly noticeable upon administration of DSS, a chemical that disrupts the intestinal basement membrane, triggering excessive ISC divisions and the accumulation of undifferentiated "stalled" EBs[16,23]. PRC2 knockdown leads to the synchronous and rapid activation of this stalled EB pool toward EC (Fig. 4) which is mechanistically linked to de-repression of *miR-8* (Figs. 5 and 6).

*miR-8* emerges as important regulator of EB differentiation. Antonello et al.[15] reported that *miR-8* is upregulated in late enteroblasts and mediates their differentiation to enterocytes, whereas depletion of *miR-8* is linked to retention of enteroblasts in an undifferentiated state. We have reproduced these findings (Supplementary Fig. 10) and showed that depletion of *miR-8* alleviates the differentiation defect caused by inactivation of E(z) in ISCs/EBs (Fig. 6C, D). Additionally, our ChIP-seq data directly implicate E(z)/PRC2 in the control of *miR-8* expression inasmuch as depletion of *E(z)* reduces H3K27me3 chromatin marks in the *miR-8* gene locus and elevates *miR-8* levels (Fig. 5). The effects of E(z) on *miR-8* expression in the *Drosophila* intestine are likely to be conserved in mammals. Our in silico analysis of normal small intestine and colorectal adenocarcinoma gene expression data confirms that EZH2 expression is inversely correlated with *miR-200*, the human homolog of *miR-8* (Supplementary Figs. 12 and 13).

Replicational dilution has been reported as the dominant mode for the loss of H3K27me3 marks upon depletion of PRC2 in the mouse intestine, but expression of PRC2 target genes varies according to thresholds of H3K27me3 loss[40]. A critical threshold of H3K27me3 levels is also required to maintain Hox gene silencing in the Drosophila wing: ablation of PRC2 activity results in dilution of H3K27me3 in each replication and progressive failure to maintain repression of the target gene[41,42]. We thus postulate that the PRC2-generated H3K27me3 mark in the *miR-8* locus is maintained in ISCs but is progressively lost during differentiation of EBs to ECs following endoreplication. The hypothesis of *miR-8* locus-specific threshold of H3K27me3 marks is partly

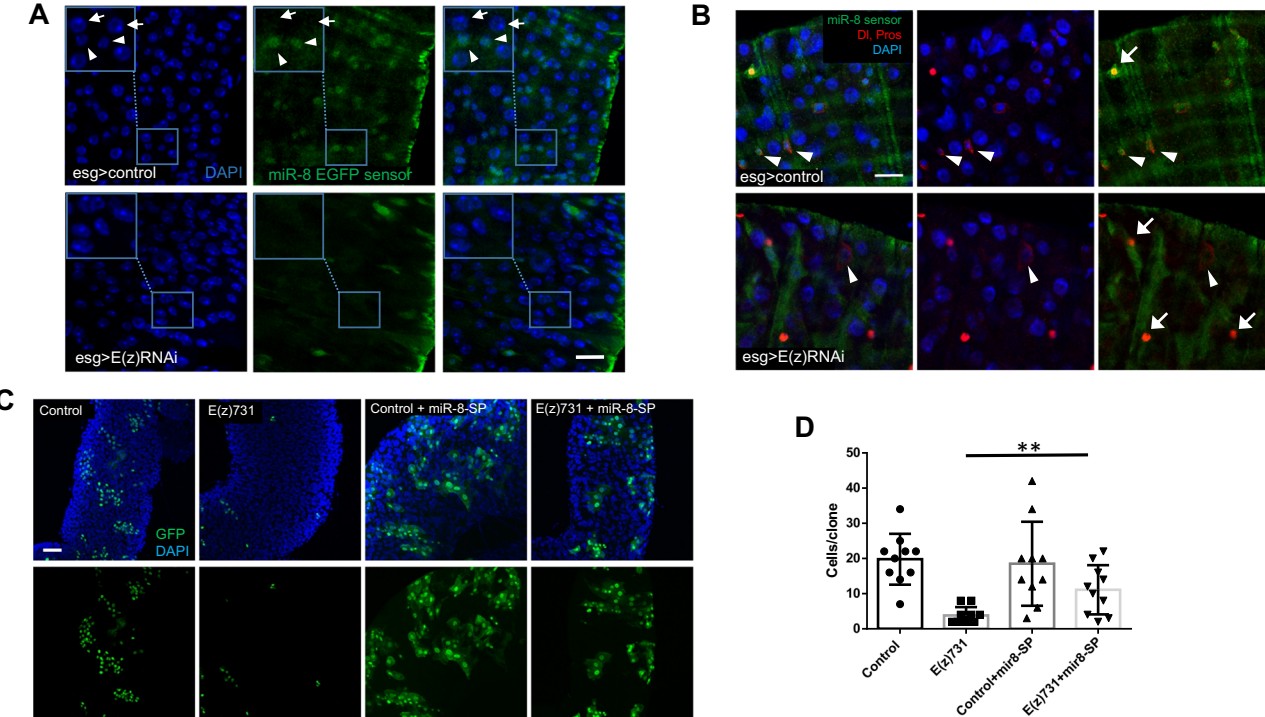

**Fig. 6 | The microRNA *miR-8* is a functional target of E(z)/PRC2. A** In control flies fed with sucrose, *miR-8*-EGFP-sensor expression is detected mainly in cells with small nuclei (arrowheads) but is absent from cells with large nuclei (arrows), suggesting low *miR-8* levels in ISCs/EBs and EEs and higher expression in committed EBs and ECs. Flies induced to express E(z)-RNAi for 7 days by an esg-Gal4 driver which does not express GFP (esg-Gal4, tubGal80ts) have decreased number of *miR-8*-sensor positive cells. Note the high levels of sensor GFP in the visceral muscle layer surrounding the gut, suggesting absence of *mir-8* expression in those cells. Scale bar; 25 μm. **B** Dual immunostaining of midguts from control or E(z)-RNAi flies with Dl (arrowheads) and Pros (arrows) antibodies. Lower expression of the *mir-8*-sensor was seen in ISCs of flies induced to express E(z)-RNAi by an esg-Gal4 driver which does not express GFP (esg-Gal4, tubGal80ts). In addition to the loss of *miR-8*-sensor-GFP, note the increased diameter and lower levels of Delta in a remaining ISC upon 14 days of E(z)-RNAi induction (arrowhead in lower panel). Scale bar; 25 μm. **C** The growth defects in E(z)[731] mutant clones are partly alleviated by depletion of *miR-8*. Depletion of *miR-8* by UAS-*miR-8*-sponge leads to expansion of GFP+ E(z)[731] MARCM clones visualized 10 days after heat shock induction. Scale bar; 50 μm. **D** Quantification of the cell numbers per clone of the indicated genotypes. Results shown are the means ± SD of *n* = 10 clones for each genotype. E(z)[731] mutant clones with depletion of *miR-8* have a greater number of cells than E(z)[731] mutant clones alone. Two-tailed, unpaired *t*-test: *p* = 0.0059.

supported by the observation that high levels of H3K27me3 in ISCs and EBs are associated with low *miR-8* activity (high *miR-8* sensor EGFP), but *miR-8* is found upregulated in ECs irrespective of the H3K27me3 levels expressed in these cells (Supplementary Fig. 11). The proposed putative link between thresholds of H3K27me3 marks and *miR-8* regulation is also aligned with our data causally linking E(z) depletion to high *miR-8* levels (Fig. 5A) and accelerated generation of enterocytes (Fig. 4). The fact that *miR-8* and Escargot have opposite effects in the control of polyploidy/diploidy[15] is also in agreement with the enlargement of the ISC nuclei and reduction of Escargot protein observed after loss of PRC2 (Supplementary Figs. 4A, C and 8B).

Whereas we have focused on *miR-8* as a crucial target of PRC2 in the adult intestine, we expect that this highly pleiotropic epigenetic modifier will most likely have additional targets relevant to ISC differentiation. It is in fact of interest that *Psc* and *Ph-p*, components of the PRC1 complex that binds H3K27me3-modified chromatin and establishes H2AK119 ubiquitylation, are targets of PRC2 (Supplementary Data 1 and Fig. 5C). It is also notable that genes encoding factors reported to establish or consolidate the EB state are also detectably methylated in our dataset (Supplementary Data 2). These include *klu*, *zfh2*, *Sox21a*, *GATAe* and *Pdm1* (*nub*), five transcription factors that mediate the inhibition of proliferation and initiation of endoreplication leading to the gradual differentiation of EBs to ECs[16,43–45]. Together with *miR-8*, which represses Esg, a gatekeeper of diploidy[15], H3K27-trimethylation of these genes would ensure their silencing in the ISCs, which would in turn prevent ISC entry into endoreplication and precocious differentiation (Fig. 8).

Lineage commitment in the *Drosophila* intestine is known to be controlled by Delta (Dl)-Notch whereby high levels of Dl expression in ISCs activate Notch in the EB daughter cell, triggering its commitment toward the enterocyte lineage[13,14,46–48]. In contrast, loss of Notch signaling leads to expansion of Dl+ ISCs and post-mitotic entero-endocrine cells[36]. Therefore, loss of Notch and loss of E(z) have opposite effects on stemness. Data presented herein demonstrate that the stemness defect caused by loss of E(z) function is remedied by RNAi-mediated depletion of Notch (Fig. 7), suggesting that E(z) is required for stem cell self-renewal by inhibiting spurious Notch activity in these cells. Although previous studies have indicated a similar antagonistic relationship between Notch and PRC2 in cell fate reprogramming in mammalian cells[49–51], the mechanism underpinning this interaction remains elusive.

We speculate that immediately after ISC division, Notch signaling removes repressive histone marks placed by E(z) on early differentiation markers, like *klu* (which suppresses the mitotic cycle) and *zfh2*, through the reported Notch-mediated induction of the K27me3 demethylase Jmjd3 (dUtx in Drosophila)[49]. Demethylation may act in concert with damage cues (e.g., Jak/STAT and Dpp) to enable the expression of later differentiation markers, like *Sox21a*, *miR-8* and *pdm1*, leading to the transition of late EBs to ECs. During this later stage, endoreplication may further dilute the H3K27me3 mark and consolidate the EC fate. In line with this hypothesis, human neural progenitor cells depleted of Jmjd3/dUtx display increased H3K27me3 content at Notch target genes and failure to differentiate into neurons and glia[52]. Another possible but not exclusive scenario is that the

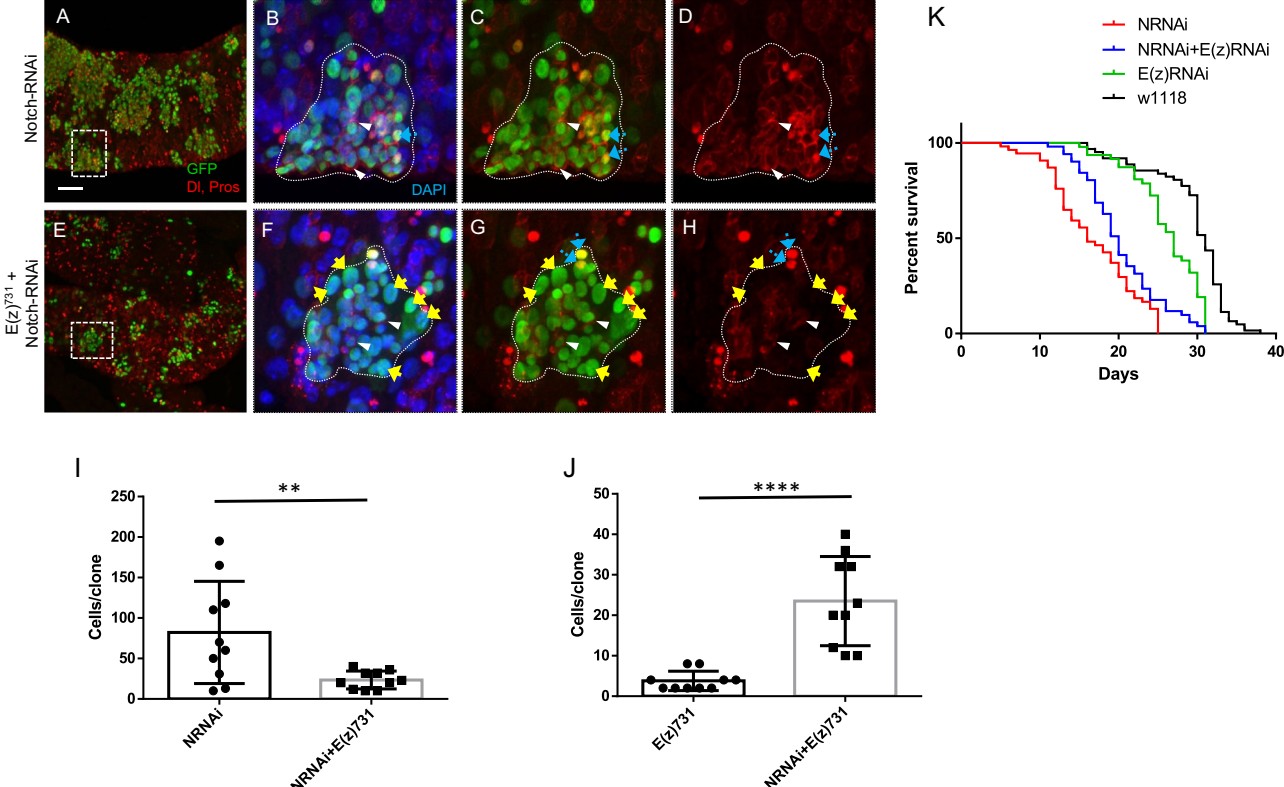

**Fig. 7 | Notch is required for the loss of stemness induced by E(z) depletion.** **A–H** The RNAi-mediated knock-down of Notch leads to accumulation of Dl[+] and Pros[+] cells in MARCM control clones 10 days after heat-shock (**A–D**). *E(z)*[731] mutant clones depleted of Notch (**E–H**) proliferate to a lesser extent than Notch-RNAi alone, and also accumulate Dl[+] and Pros[+] cells. Compared to *E(z)*[731] null MARCM clones that hardly proliferate (Figs. 3 and 6), the simultaneous depletion of Notch displays a dramatically elevated number of cells per clone (**E**). Scale bar; 25 μm; arrowheads show representative Dl[+] cells; light blue dotted arrows show representative Pros[+] cells and yellow arrows indicate large Dl[−] and Pros[−] cells, most likely representing enterocytes. **I**, **J** Quantification of the average cell numbers per clone

of the indicated genotypes. Results shown are the means ± SD of *n* = 10 clones for each genotype. *E(z)*[731] mutant clones depleted of Notch (**E–H**) grow to a lesser extent than those of Notch-RNAi alone (Two-tailed, unpaired *t*-test: *p* = 0.0059) (**I**), but significantly more than *E(z)*[731] null clones (**J**, see also Figs. 3 and 6). (Two-tailed, unpaired *t*-test: *p* < 0.0001). **K** The reduced survival of flies bearing Notch RNAi in ISCs/EBs is partly alleviated upon simultaneous E(z) knock-down. Survival curves of the indicated lines crossed to esg-Gal4 driver are shown for the indicated genotypes (*p* = 0.0053, log-rank (Mantel–Cox) test, *n* = 55 for N-RNAi, *n* = 51 for E(z)-RNAi, *n* = 47 for Notch plus E(z)-RNAi and *n* = 62 for *w*[1118] flies, pooled from three independent experiments).

Su(H)/RBPJ transcription factor that mediates Notch signaling may gain access to Notch target gene promoters upon replicational dilution of H3K27me3 marks. Of note, our in silico analysis of the *Drosophila miR-8* locus indicates the presence of four potential Su(H) binding sites (Supplementary Fig. 14G), suggesting that *miR-8* is a Notch target.

On the basis of the findings reported herein, we propose that whereas Notch signaling is required in EBs to induce differentiation to ECs via upregulation of *miR-8* and subsequent downregulation of its target Esg, E(z)/PRC2 is required in ISCs to maintain potential Notch target genes, such as *miR-8*, in a repressive chromatin state and thus prevent precocious differentiation due to basal Notch signaling (Fig. 8). These interactions may have implications for understanding plasticity in aging. H3K27me3 is upregulated in gut progenitors of old flies (Supplementary Figs. 9A and 13B), suggesting that the overall activity of PRC2 is increased and/or the activity of the dUtx K27 demethylase is decreased upon aging. The H3K27me3 upregulation is consistent with levels of *miR-8* being lower in aged gut progenitors cells (Supplementary Figs. 9A and 13B), supporting the notion that defects in differentiation have a primary role in aging of the intestine[53].

In summary, our study demonstrates that the evolutionarily conserved E(z)/PRC2 epigenetic modifier and the microRNA *miR-8* act in concert with stemness and differentiation transcriptional circuits governed by Esg and Notch to balance cell fate decisions and safeguard intestinal homeostasis.

## Methods
### Drosophila husbandry and genetics
Information on Drosophila genes and lines used in this study is available in Flybase (http://flybase.bio.indiana.edu). All cultures were kept on standard yeast-cornmeal diet under controlled humidity and light laboratory conditions at the IMBB insect facility. Mated females rather than males were used because of the larger midgut size in females and because the dynamics of damage-induced intestinal regeneration are different in females and males, and mated and virgin females[54,55]. No institutional ethical approval is required for experiments with *Drosophila melanogaster*.

For the RNAi experiments, esg-Gal4, UAS-GFP, tubP-Gal80ts flies[13] were crossed to flies carrying the UAS-RNAi construct or to control lines *w*[1118], UAS-lacZ or UAS-wRNAi, as indicated in the figure legends. The crosses were kept at 18 or 25 °C and F1 flies shifted to 29 °C for at least 2 days to allow expression of the transgenes. The following lines were acquired from VDRC: *w*[1118] (V60000), *E(z)*-RNAi (V27645, V27646, V107072), *Su(z)*12-RNAi (V42423) and *esc*-RNAi (V5690). Flies were fed with sucrose (5%) or DSS (3%, MP Biomedicals) in the presence of 5% sucrose in empty vials containing soaked filter paper prior to dissection. Progenitor cells were quantified by counting GFP-positive cells. GFP was assessed in a qualitative manner, no attempt was made to measure GFP intensity which might be altered due to competition for the Gal4 in RNAi lines. UAS-lacZ (Bloomington 3955) and UAS-wRNAi

(Bloomington 35573) were crossed to esg-Gal4 and used as additional controls to w[1118].

For the MARCM system, virgin females of the genotype yw, hsFLP, tubP-Gal4, UAS-GFP/FM7; tubP-GAL80 FRT2A/TM6B were crossed to either w; FRT2A/TM3 for control clones or E(z)[731] FRT2A/TM6C (Bloomington 24470) to generate E(z) null mutant clones. Su(z)12[4] allele (Bloomington 24469) was used to create Su(z)12 mutant clones. Mated female flies, 4 to 7-day-old, were heat-shocked for 40 min at 37 °C to induce somatic recombination. Clones were observed 10 days after induction.

For the G-TRACE experiments UAS-Flp, UAS-RFP, Ubi-FRT-STOP-FRT-nEGFP (Bloomington 28281) flies carrying an esg-Gal4, tubP-GAL80ts chromosome were crossed to either w[1118] (control) or E(z)-RNAi. Four to 7 days old flies bearing all the above transgenes were shifted to 29 °C in empty vials containing soaked filter paper in 5% Sucrose or 3% DSS in the presence of sucrose for 36 h. Flies were then shifted back to regular food for 4 days before dissection and imaging at indicated times.

UAS-mir-8[56], UAS-mir-8-sponge[57] and miR-8-sensor[31] were kindly provided by Maria Dominguez. The UAS-Notch-RNAi line was obtained from Bloomington (BL7078).

## Immunofluorescence staining and image analysis

Adult midguts were dissected into 1× phosphate-buffered saline (PBS) and fixed in 4% formaldehyde for 30 min at room temperature. Delta staining was performed following a methanol fixation protocol. Samples were then washed 3 × 10 min in 1× PBS with 0.2% Triton X-100 (PT) and incubated in blocking solution (PBT: PT with 0.5% bovine serum albumin) for 30 min at room temperature. Samples were incubated with primary antibodies diluted in PBT overnight at 4 °C, washed 3 × 10 min at room temperature in PT, incubated with secondary antibodies diluted in PBT at room temperature for 1 h, washed 3 × 10 min with PT, and mounted in Drop-n-Stain EverBrite™ Mounting Medium with DAPI (Biotium).

The following primary antibodies were used: phospho-Histone H3 Ser10 (pH3) 1:1500 (Merck Millipore, 06-570), mouse anti-Delta (C594.9B, 1:50 DSHB), mouse anti-Prospero (MR1A, 1:50, DSHB), mouse anti-Pdm1 (NUB 2D4, 1:50 DSHB), rat anti-Esg (gift from Claude Desplan) and rabbit anti-H3K27me3 1:1500 (Merck Millipore 07-449). Secondary antibodies (Biotium and Invitrogen) used diluted (1:1000) in PBT. The ApopTag Red In Situ Apoptosis Detection Kit (S7165; Sigma-Aldrich) was used for apoptosis detection.

Images were taken from the posterior R4 part of the midguts with a Leica SP8 confocal microscope system (IMBB confocal facility). Image analysis was performed using the Leica LAS X software (3.7.2.22383). In the G-TRACE experiment newly-produced ECs were discerned because of their bigger size and the expression of high levels of EGFP and low or no RFP (whereas progenitors are strong RFP/weak EGFP and older ECs are negative for both markers). Quantifications were performed using the StarDist 2D ImageJ/FiJi (1.53t) plugin. Graphs, statistical analysis, and survival curves produced in GraphPad Prism 6. Significance was calculated using unpaired two-tailed Student's t test.

## FACS-sorting and RT-qPCR

The protocol for gut dissociation and FACS sorting was adapted from Dutta et al.[63]. Briefly, 100 midguts from esg-Gal4, UAS-GFP, tubP-GAL80ts flies crossed to w[1118] or E(z)-RNAi dissected in ice-cold PBS and immediately digested with elastase solution (1 mg/ml, Sigma-Aldrich). Dissociated intestinal cells were centrifuged mildly, filtered and sorted by a FACSAria III cell sorter (BD Biosciences) into lysis buffer (QIAzol, Qiagen). Total RNA including micro RNAs was extracted using the miRNeasy Micro Kit (Qiagen) and reversed transcribed using Super-Script III Reverse Transcriptase (Invitrogen) for coding genes or the

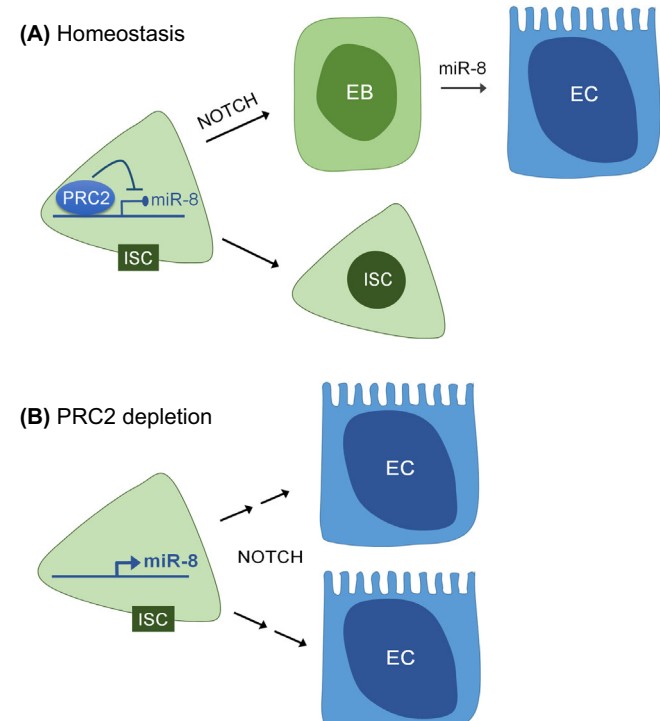

**Fig. 8 | Proposed model of PRC2-mediated regulation of intestinal stem cell (ISC) fate in Drosophila.** Based on the results reported herein, we propose a dynamic relationship between E(z), miR-8 and Notch signaling in controlling stemness vs. differentiation of ISCs. Our data suggest that E(z)/PRC2 is required in ISCs to maintain Notch target genes, such as miR-8, in a repressive chromatin state, functioning as a brake to precocious differentiation caused by basal Notch signaling (**A**). Disruption of E(z)/PRC2 function specifically in ISCs leads to loss of stemness and differentiation toward the EC lineage associated with Notch-mediated upregulation of miR-8 and, consequently, reduced levels of the miR-8 target Escargot (**B**).

miRCURY LNA kit (Qiagen) for miR-8. cDNA was then mixed with a SYBR master-mix (KAPA SYBR Green fast Master PCR Kit (Roche) or miRCURY SYBR Green (Qiagen) respectively. The qPCR reactions were performed with a Bio-Rad 1000-series thermal cycler. rp49 or ribosomal 2S RNA primers were used for normalization, and relative expression levels were calculated with the comparative CT method, providing similar results. Primers for qRT-PCR were designed for coding genes using FlyPrimerBank (https://www.flyrnai.org/flyprimerbank) as follows: for E(z): forward AGTGGAAGAGGC GTGTCAAG; reverse TGTGCTCGTCCCAGTTTCTTAT, for l(2)gl: forward TGACGACCACCACTTTGTGT; reverse CTTTTGTTGGTTTGCG TGCTA, for Psc forward ATGACGCCAGAATCGAAAGCA; reverse CGACTTTTGTTGTGTGTGAGC and for ph-p forward AGTAGCGTG CCCTTCTCAGT; reverse CCGGCTTAGCTGTCTGAAAGA. For the amplification of miR-8 commercially available primers from Qiagen were used (MIMAT0000113 for dme-miR-8-3p and MIMAT0020791 for dme-miR-8-5p).

## ChIP-Seq and data analysis

Young (2–3 days old) mated females were moved to 29 °C for 4 days to express E(z)-RNAi and then dissected; 9 × 10⁴ and 7 × 10⁴ esgGal4 > GFP-positive cells were isolated from control and E(z)-RNAi lines respectively by FACS-sorting, snap-frozen in liquid nitrogen and shipped unfixed to Active Motif to perform low cell number ChIP-Seq. Chromatin was precipitated with an antibody against H3K27me3, to determine the presence of the repressive histone mark across the

genome and regions with decreased occupancy upon E(z)-RNAi. Immunoprecipitated and input DNAs were processed into Illumina sequencing libraries, using NextSeq 500.

The 75 nt reads generated, were aligned to the Drosophila genome using the BWA algorithm (default settings). Duplicate reads were removed and only uniquely mapped reads (mapping quality ≥25) were used for further analysis. Alignments were extended in silico at their 3'-ends to a length of 200 bp (using Active Motif software). Peak locations were determined using SICER algorithm. To associate peaks signals to genes, the GeneMargin, distance upstream and downstream of a gene that determines whether an interval is associated with that gene, was set up to 500 bp upstream or downstream of a gene. SICER-df was used to identify regions with decreased H3K27me3 in E(z)-RNAi progenitor cells.

### In silico analysis of public human data

Counts from TCGA-COAD and GTEx healthy colon after uniform re-processing were obtained from the Recount3 project[58] through the Recount3 R package (https://bioconductor.org/packages/release/bioc/html/recount3.html). Genes with less than 10 counts in less than 10% of samples in both categories were removed. Data were normalized using Median of Ratios and Differential Expression Analysis was performed with DESeq2 (ref. 59). Log2 Fold Changes were shrunk using the adaptive Student's t prior shrinkage estimator and significance was re-computed using a log fold change threshold of 0.5 (ref. 60). p values were corrected using the Benjamini Hochberg method[61]. To visualize expression, counts were subjected to a non-blind Variance Stabilizing Transformation[62–64].

Single cell RNAseq data for EZH2 was retrieved from the Human Protein Atlas database, https://www.proteinatlas.org/.

### Reporting summary

Further information on research design is available in the Nature Portfolio Reporting Summary linked to this article.

## Data availability

Data are available within the article and Supplementary Information. ChIP-seq data generated in this study are available in the National Center for Biotechnology Information database under BioProject ID PRJNA934770. Source data are provided with this paper.

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

## Acknowledgements

We thank Sotirios Kampranis, George Apidianakis, Chrysoula Pitsouli, Maria Monastirioti, Eva Zacharioudaki, Eleftheria Ledaki, Amalia Riga, Katerina Kadianaki and Athanasia Stamelou for providing help and advice at various stages of the project. We thank Zacharenia Vlata from the IMBB FACS sorting facility for assistance with FACS-sorting. We are grateful to Paul Labhart from Active Motif, Maria Tsagiopoulou, Nikos Papakonstantinou & Fotis Psomopoulos and Vasiliki Theodorou for help with ChIP-seq data analysis. We also thank BDSC and VDRC for providing Drosophila stocks, DSHB for antibodies and Flybase for genetic information. This study was supported by an EU-FP7 REGPOT grant (Contract Number 285948) to A.G.E., funding by Latsis Foundation, IKY-Siemens, Fondation Santé and the Hellenic Foundation for Research & Innovation (Grant No. 1561) to A.G.E. and Z.V, and the PRO-sCAP grant to A.G.E supported by Greece 2.0 - National Recovery and Resilience Plan (Grant No. TAEDR-0541976).

## Author contributions

Z.V. and A.G.E. conceived the project, designed and supervised experiments. Z.V. and V.F. performed experiments and analyzed data. N.K., I.S.V. and A.G.E. performed human data analysis. C.D. supervised Drosophila genetic experiments and critically reviewed the manuscript. Z.V. and A.G.E. wrote the manuscript. All authors read and approved the manuscript as submitted.

## Competing interests

The authors declare no competing interests.
