## [Peer Review File · Nature Communications]

Polycomb-mediated silencing of *miR-8* is required for maintenance of intestinal stemness in *Drosophila melanogaster*REVIEWER COMMENTS

Reviewer #1 (Remarks to the Author):

The ability of adult intestinal stem cells to balance self-renewal and differentiation remains incompletely understood. Here the authors describe that an epigenetic-based mechanism involving the Polycomb Repressive Complex 2 (PRC2) is required to maintain the intestinal stem cell (ISC) pool in adult *Drosophila* via the regulation of the mir-200 orthologue and Notch.

In earlier studies, Koppens et al. (2016) and Chiacchiera et al. (2016), have studied the impact of conditional deletion of EED in mice to disrupt PRC2 in the intestine. They found that loss of PRC2 function led to increased secretory lineage cells in the entire intestinal epithelial population.

Additionally, they identified Atonal homolog-1 (Atoh1), a transcription factor required for secretory cell commitment downstream of Notch signalling, as a relevant PRC2 target. However, these studies did not investigate the role of PRC2 in specific intestinal cell populations, including ISCs, or provide a mechanism that may underlie intestinal stem cell plasticity and response to challenges.

The paper presents a well-structured and logically sound model that fits the data well. The proposed approach is attractive due to its simplicity and clear logic, and the results are solid and phenotypic consequences are also explored. Also, exploring a possible link with the ageing intestine, which would be relatively easy, may expand the implications of the findings for future research and practical applications and give additional interesting connections.

Epigenetic marks are often associated with “memory” and the stability of gene expression programmes over time. This study implies a possible dynamic role for the E(z) and PRC2 in the plasticity of stem cells to balance self-renewal and differentiation and repair, with interesting implications.

Still, some gaps in the study should be addressed. For example, the limitations and assumptions of the model are not fully discussed —Is E(z) acting in the ISCs or the EB, and how is E(z) or H3K27me3 regulated? Is E(z) in ISCs regulating mir-8 or the EBs with Notch?

Addressing these gaps would further strengthen the study. Below are my comments and minor points.

- Line 48. Please add a citation to the statement, “PcGs are now widely recognised for their role in.....remodelling and repression of gene expression”

- Line 221. The precise age of the flies for ChIP-seq is not clear.

- In figure legend

“Posterior midguts from a 7-day-old adult female fly induced to express E(z)-RNAi for 4 days in ISCs/EB fed with sucrose (C) or DSS (D). Note the absence of the DSS-induced increase in the GFP+ cells”

This needs some clarification: Are flies 3 days old shifted to induce expression of E(Z)-RNAi for 2 days and then fed sucrose or sucrose +DSS for additional two days, and guts dissected at day 7? Is that correct? Or is the DSS treatment of four days along with E(Z)-RNAi?

- Are the guts shown all females and mated? This should be stated clearly in the methods with a statement that the reason to use only females for the less knowledgeable reader of the non-*Drosophila* expert is that the dynamics of repair are different in females and males, and mated and virgins.

- figure 4. It appears surprising that in control guts, the number of renewing ECs is like that of DSS-treated guts; I would have expected an increase or more differences in the number of ECs renewing,

particularly on day 1, as suggested by the image in fig. 4E. Could the authors comment on that?

- The images in this figure 4 are very convincing of the transformation towards differentiation. The DSS treatment seems mild, yet the differences with E(z) knockdown are evident, and the conversion of ISCs to EBs and ECs very clear. In the previous figures, some E(z) RNAi-expressing cells were still with the morphology of EBs, but figure 4, which might show guts of longer exposure to RNAi against E(z), is very convincing of the effect of E(z) proposed in the model.
- A magnification of figure 5A will help to see the localisation of the H3K27me3 marks. Some H3K27me3 are labelling in the GFP-negative small nuclei cells, which may be the Prospero-positive cells, and in some GFP-positive cells, some, judging by their size, appear to be EBs. To clarify that the E(z) and H3K27me3 marks are also (or predominantly) in ISCs to prevent spurious Notch signalling, would it be possible to co-staining intestines with anti-Delta, or other ISC-markers and H3K27me3?
- Supplementary Figure 5. The images suggest that all intestinal cells have high H3K27me3. Would not the EB have some depletion of E(z)? Comparing the staining in figure 5 and supple. Figure 5, the proposed depletion/dilution of the silencing mark in polyploid cells is unclear.
- In the clones of E(z), levels of Delta seem lower; however, in supplementary Fig. 2 the staining of Delta in large cells does not suggest downregulation of Delta. Could the authors comment on this and speculate more about how E(z)-mediated maintenance of stemness is integrated with local repair and differentiation and dynamic Delta-Notch activation.
- The model explains the consequence of E(z) loss. However, it would be interesting to speculate on how the dynamic E(z) PCR2 regulation enables the activation of mir-8 during homeostasis and local responses to injury. For example, the model shows how the epigenetic marks seemed up-regulated in response to DSS treatment. The authors also should try to more convincingly illustrate the proposed depletion/dilution of the epigenetic marks and its association with polyploidy or expression of the mir-8 sensor and complementarity with the epigenetic marks. Specifically, show the cells with high mir-8 sensor higher H3K27me3?

These interesting interactions may have implications for understanding plasticity in ageing. Do the levels of H3K27me3 change in the intestines of older flies? Other authors have shown that the defect in repair is not due to a lack of ISC mitosis but differentiation failure, which hints at a role for aberrant E(z) mediated regulation and failure to activate mir-8 in older flies...

Reviewer #2 (Remarks to the Author):

This work studies the Polycomb components, mainly the E(z), in the *Drosophila* midgut intestinal stem cell lineage. The authors used genetics, cell biology and genomics tools and they claimed that E(z) is required for proper ISC differentiation. Molecularly, this regulation is through a microRNA target mir-8. Overall, this kind of studies would be useful for the field to understand better how chromatin regulators contribute to adult stem cell lineages. However, there are so many issues with the studies presented in this work, both technically and biologically. Technically, there is an issue of the control for RNAi (see

below) and many results were shown as single images without quantifications. In addition, many loss-of-function experiments were done at one time point, which cannot give out a comprehensive picture biologically. Overall, with the current status, this work is too preliminary to be considered for publication. More rigorous and better controlled experiments are required to move forward.

1. Figure 1, 2, 5, 4, 6, S1C, S2, S3, S5, according to the figure legends, “adult female flies expressing *esg-Gal4*, *tubGal80ts*, *UAS-GFP (esgts>GFP)*” which were the progenies from the cross to *w1118* strain was used as the RNAi control. This is not an appropriate control. The right control should utilize the same driver combination with a non-specific RNAi, such as a *lacZ* or *mCherry* RNAi. This way the driver drives both *UAS-GFP* and *UAS-control RNAi* or *UAS-PcG RNAi*, so that any difference of GFP is less likely due to competition for the *Gal4*. This is important for data interpretation as GFP signals have been used as a quantification method, and the GFP signals are less in the RNAi lines than in the control line.

2. Quite some quantifications used “representative images from posterior midguts”, what does it mean? How many representative images per midgut?

3. Fig. S3: With the sucrose, the apoptotic cells decrease significantly in *E(z)*-RNAi strain, any explanation for that? Still, this experiment needs to use *UAS-control RNAi* instead of crossing to *w1118*. Maybe this question can be addressed with the new control experiments.

4. All quantifications should be accompanied with supplemental tables.

5. Scale bars need to be shown for all figure panels, currently some have but many do not.

6. “MARCM analysis of *Su(z)12* mutated clones produced similar results (data not shown).” These data should be shown.

7. The MARCM results were analyzed “at 12 days after induction of recombination” therefore it is just one snapshot after inducing mitotic clones, a more careful time-course should be done to get an idea of how *E(z)* mutant ISCs biases differentiation toward the EC fate. Moreover, the Fig.3F showed that Delta-positive cells decline in *E(z)* mutant clones, image showing this should be provided as well. On the other hand, in Fig. S4, images were shown (positive with *Pdm1* and negative with *esg*) but no quantification is provided. In summary, for the stem cell differentiation assay using different markers, images should be accompanied with quantification, and with the clonal analysis, a time-course series of data including both images and quantification should be provided to make a rigorous assay.

8. Fig.4 shows G-TRACE results, which is very confusing. First of all, why using the same GFP for both real-time and lineage tracer. Even though the localization difference may be able to separate these two populations, based on the images it is hard to distinguish without other cellular markers. Since the nGFP signal is stronger, the cytoplasmic GFP signal became less detectable in the presence of nGFP, such as in Fig.4C-D, G-H, I-P. Why not using a difference marker, such as *mCherry*, for the real-time expression, as used in many other G-TRACE experiments done in different systems in *Drosophila*. Furthermore, the perdurance of GFP could make the real-time label beyond just “progenitor cells (ISCs and EBs)”.

9. Line 74-75: “intestinal stem cells (ISCs), which are the only mitotic cells in this tissue” needs revision as recent studies showed *ee* cells can also divide.

10. “miR-8 precursor (3p and 5p)”, a cartoon illustrating their localization will be helpful.

11. “we observed decreased ISC mitotic rates after miR-8 overexpression, and over-accumulation of undifferentiated cells after depletion of miR-8, especially upon DSS treatment (Suppl. Figure 5B).” Where is the quantification for these statements?

12. Fig.6, here other than the microRNA sensor (EGFP), is GFP signal still present from the *esg>E(z)* RNAi as shown in previous experiments and figures? Again, for both major conclusions regarding *mir8* expression (Fig. 6A-B) and its functional relevance (Fig. 6C-F), only one image is shown for each without

any quantifications.

13. Fig.7, same issue that just one image is shown without any quantifications. Also, in this figure, no labeling was indicated and no scale bar was included. And only one time point was shown.

Reviewer #3 (Remarks to the Author):

In the manuscript entitled "Polycomb-mediated silencing of miR-8 is required for maintenance of intestinal stemness in *Drosophila melanogaster*" authors report that the Polycomb Repressive Complex 2 (PRC2) is required for maintenance of the intestinal stem cell (ISC) pool in adult *Drosophila*.

To elucidate the role in vivo of E(z)/PRC2 in ISCs fate decisions and lineage progression they investigated its role under normal and pathological conditions. Authors clearly showed that E(z) depletion affected ISC proliferation suggesting that E(z)/PRC2 is required for the maintenance of a functional ISC pool that is critical for regenerative capacity in the *Drosophila* midgut.

Authors showed that knocked down E(z) in intestinal stem cells and enteroblasts (ISCs/EBs) by activating RNAi expressed under the control of the conditional, temperature-sensitive *esg-Gal4*, *UAS-GFP*, *tub-GAL80ts* induces precocious differentiation into ECs.

Veneti et al. showed that loss of PRC2 activity in ISCs by RNAi, results in loss of stemness and precocious differentiation of enteroblasts to enterocytes under normal and pathological conditions.

Performing H3K27me3 ChIP-seq experiments authors identified direct targets of E(z)/PRC2 in ISCs/EBs.

Among those targets that displayed the largest reductions in H3K27me3 upon E(z) knockdown they identified *l(2)gl*, *ph-p*, *Psc* and *miR-8* loci. Due to *mir8* involvement in the control of EB differentiation to ECs in the *Drosophila* intestine, authors focused their attention on *mir-8*. Of note, *miR-8* overexpression promotes enteroblast to enterocyte transition

At the same extent of the E(z) knock-down. They concluded that *miR-8* is a functional target of E(z)/PRC2-mediated H3K27me3 in intestinal progenitors playing a critical role in controlling stemness versus differentiation of ISCs.

Then, starting from the observation that Notch signalling is required in EBs to induce differentiation to ECs, they studied the possible link with E(z)/PRC2. Authors claimed that loss of Notch and loss of E(z) have opposite effects on stemness. Stemness defect caused by loss of E(z) function is remedied by RNAi-mediated depletion of Notch. Based on these findings the authors propose that E(z) is required for stem cell self-renewal by inhibiting spurious Notch activity.

Overall, the presented work is of good quality and in some part convincing but fell short in extending the current molecular knowledge that links Notch signalling with PRC2. It has been already reported that loss of PRC2 (EED KO) affects cell plasticity at the crypt bottom by inducing differentiation towards the secretory lineage downstream of Notch activation. Thus, the link between PRC2 and the Notch signaling has been previously well documented.

In my opinion no additional mechanistic insights are provided to extend this knowledge.

NCOMMS-23-08936: Polycomb-mediated silencing of *miR-8* is required for maintenance of intestinal stemness in *Drosophila melanogaster*

Zoe Veneti, Virginia Fasoulaki, Nikolaos Kalavros, Ioannis S. Vlachos, Christos Delidakis & Aristides G. Eliopoulos.

REBUTTAL LETTER

REVIEWER COMMENTS

Reviewer #1

REVIEWER'S OVERALL COMMENTS: *The ability of adult intestinal stem cells to balance self-renewal and differentiation remains incompletely understood. Here the authors describe that an epigenetic-based mechanism involving the Polycomb Repressive Complex 2 (PRC2) is required to maintain the intestinal stem cell (ISC) pool in adult Drosophila via the regulation of the mir-200 orthologue and Notch.*

In earlier studies, Koppens et al. (2016) and Chiacchiera et al. (2016), have studied the impact of conditional deletion of EED in mice to disrupt PRC2 in the intestine. They found that loss of PRC2 function led to increased secretory lineage cells in the entire intestinal epithelial population. Additionally, they identified Atonal homolog-1 (Atoh1), a transcription factor required for secretory cell commitment downstream of Notch signalling, as a relevant PRC2 target. However, these studies did not investigate the role of PRC2 in specific intestinal cell populations, including ISCs, or provide a mechanism that may underlie intestinal stem cell plasticity and response to challenges.

The paper presents a well-structured and logically sound model that fits the data well. The proposed approach is attractive due to its simplicity and clear logic, and the results are solid and phenotypic consequences are also explored. Also, exploring a possible link with the ageing intestine, which would be relatively easy, may expand the implications of the findings for future research and practical applications and give additional interesting connections. Epigenetic marks are often associated with "memory" and the stability of gene expression programmes over time. This study implies a possible dynamic role for the E(z) and PRC2 in the plasticity of stem cells to balance self-renewal and differentiation and repair, with interesting implications.

Still, some gaps in the study should be addressed. For example, the limitations and assumptions of the model are not fully discussed —Is E(z) acting in the ISCs or the EB, and how is E(z) or H3K27me3 regulated? Is E(z) in ISCs regulating mir-8 or the EBs with Notch? Addressing these gaps would further strengthen the study. Below are my comments and minor points.

RESPONSE TO REVIEWER'S OVERALL COMMENTS: We are grateful to the reviewer for his/her positive remarks and insightful comments that enabled us to strengthen the submitted work. In

the revised manuscript we provide additional information and new data to address the reviewer's constructive critique, as detailed below:

(A) ***Is E(z) acting in the ISCs or the EB?*** We thank the reviewer for raising this important point. To determine whether E(z) acts in ISCs, EBs or both progenitors, we have performed experiments using drivers specifically targeting E(z)-RNAi in EBs ((Su(H)GBE-Gal4, UAS-GFP; tub-Gal80ts/UAS-E(z)RNAi) or ISCs (esg-Gal4, UAS-GFP; Su(H)GBE-Gal80, tub-Gal80ts/UAS-E(z)RNAi). We have found that it is the knock-down of E(z) in ISCs, but not EBs, that reproduces the effect of E(z) depletion in intestinal progenitor cells. The new data is shown in Supplementary Figure 3 and described in lines 138-143 of the revised MS. We therefore conclude that E(z) is required for the maintenance of ISC pool and regenerative capacity in the *Drosophila* midgut by predominantly acting in ISCs.

(B) ***Is E(z) in ISCs regulating mir-8 or the EBs with Notch?*** Our work, including the ISC vs EB driver-specific experiments described above, supports a model by which E(z) keeps *mir-8* repressed in ISCs, whereas Notch signaling is required for timely expression of *mir-8* in EBs and terminal differentiation. The proposed model is summarized in lines 404-408 of the Discussion and the modified Figure 8 of the revised MS.

REVIEWER'S COMMENT 1: Line 48. Please add a citation to the statement, "*PcGs are now widely recognised for their role in.....remodelling and repression of gene expression*".

RESPONSE TO REVIEWER'S COMMENT 1: Citations have been included, thank you for pointing out this omission.

REVIEWER'S COMMENT 2: Line 221. The precise age of the flies for ChIP-seq is not clear.

RESPONSE TO REVIEWER'S COMMENT 2: This information is included in the Materials and Methods of the revised MS, line 519. We thank the reviewer for pointing out this omission.

REVIEWER'S COMMENT 3: *In figure legend "Posterior midguts from a 7-day-old adult female fly induced to express E(z)-RNAi for 4 days in ISCs/EB fed with sucrose (C) or DSS (D). Note the absence of the DSS-induced increase in the GFP+ cells". This needs some clarification: Are flies 3 days old shifted to induce expression of E(Z)-RNAi for 2 days and then fed sucrose or sucrose +DSS for additional two days, and guts dissected at day 7? Is that correct? Or is the DSS treatment of four days along with E(Z)-RNAi?*

RESPONSE TO REVIEWER'S COMMENT 3: Indeed, young (3 days old) mated females were shifted to 29°C for two days and then fed with sucrose/DSS for two additional days at 29°C. We have modified the legend of Figure 1C,D to clarify this point (lines 555-558 of revised MS).

REVIEWER'S COMMENT 4: *Are the guts shown all females and mated? This should be stated clearly in the methods with a statement that the reason to use only females for the less*

knowledgeable reader of the non-Drosophila expert is that the dynamics of repair are different in females and males, and mated and virgins.

RESPONSE TO REVIEWER'S COMMENT 4: Indeed, only mated females were used in our study. In the revised MS, lines 445-448, we have included this information and the relevant statement suggested by the reviewer.

***REVIEWER'S COMMENT 5:** Figure 4. It appears surprising that in control guts, the number of renewing ECs is like that of DSS-treated guts; I would have expected an increase or more differences in the number of ECs renewing, particularly on day 1, as suggested by the image in fig. 4E. Could the authors comment on that?*

RESPONSE TO REVIEWER'S COMMENT 5: We thank the reviewer for bringing this point to our attention, requiring clarification. We would like to clarify that in the original Figure 4E, day 1 refers to 24 hrs of DSS treatment and not time of recovery. Following DSS treatment, in control guts we had indeed observed accumulation of cytoplasmic GFP+ cells that represent EBs (expressing *esg-Gal4*, UAS-cytoplasmic-GFP at the time of tissue fixation), but not of nuclear GFP+ ECs (which are enumerated in the original Fig. 4Q & R; expressing *ubi* promoter-nuclear GFP after *esg-Gal4* UAS-FLP mediated removal of a STOP cassette within the last 24 hours -or more- before fixation). This observation fully aligned with a previous study (PMID: 19128792).

To provide a more clear demonstration of this effect, we have repeated the experiment of Figure 4 using an improved G-TRACE system, where progenitor cells are marked with RFP and newly produced progeny cells with nuclear EGFP, as suggested by another reviewer. As shown in new Figure 4 and described in lines 190-206, as well as the Methods ("Drosophila husbandry and genetics") of the revised MS, the results confirm our original finding that E(z) depletion accelerates differentiation of EBs to ECs.

***REVIEWER'S COMMENT 6:** The images in this figure 4 are very convincing of the transformation towards differentiation. The DSS treatment seems mild, yet the differences with E(z) knockdown are evident, and the conversion of ISCs to EBs and ECs very clear. In the previous figures, some E(z) RNAi-expressing cells were still with the morphology of EBs, but figure 4, which might show guts of longer exposure to RNAi against E(z), is very convincing of the effect of E(z) proposed in the model.*

RESPONSE TO REVIEWER'S COMMENT 6: We thank the reviewer for his/her positive remarks. We also find the phenomenon of interest. In the revised MS we provide a new Figure 4 using an alternative G-TRACE system which confirms the model proposed in the initial submission.

***REVIEWER'S COMMENT 7:** A magnification of figure 5A will help to see the localisation of the H3K27me3 marks. Some H3K27me3 are labelling in the GFP-negative small nuclei cells, which may be the Prospero-positive cells, and in some GFP-positive cells, some, judging by their size,*

appear to be EBs. To clarify that the E(z) and H3K27me3 marks are also (or predominantly) in ISCs to prevent spurious Notch signalling, would it be possible to co-staining intestines with anti-Delta, or other ISC-markers and H3K27me3?

RESPONSE TO REVIEWER'S COMMENT 7: Please see our response to reviewer's comment 10.

REVIEWER'S COMMENT 8: Supplementary Figure 5. The images suggest that all intestinal cells have high H3K27me3. Would not the EB have some depletion of E(z)? Comparing the staining in figure 5 and suppl. Figure 5, the proposed depletion/dilution of the silencing mark in polyploid cells is unclear.

RESPONSE TO REVIEWER'S COMMENT 8: Please see our response to reviewer's comment 10.

REVIEWER'S COMMENT 9: In the clones of E(z), levels of Delta seem lower; however, in supplementary Fig. 2 the staining of Delta in large cells does not suggest downregulation of Delta. Could the authors comment on this and speculate more about how E(z)-mediated maintenance of stemness is integrated with local repair and differentiation and dynamic Delta-Notch activation.

RESPONSE TO REVIEWER'S COMMENT 9: In the E(z)-null clone shown in Figure 3 there is no detectable DI, indicating absence of a stem cell (likely differentiated). We have now performed a more detailed analysis at different time-points and have quantified the number of DI cells in E(z)⁷³¹ versus control wt clones. As shown in Suppl Fig. 6, the DI-positive cells are severely reduced in the mutant background. Whereas we can confidently score DI presence, we cannot conclude safely about DI levels, because DI immunoreactivity is particularly sensitive to fixation conditions. The high levels of DI in the aberrant stem cells seen in Suppl Fig. 2 (now Suppl Fig. 4) could in part be due to such technical reasons, but may also be due to the more modest downregulation of E(z) activity by RNAi (in the present case), compared to the severe knockout in homozygous mutant clones (Figure 3). We have speculated about how E(z) may be intersecting with Notch signalling in local repair and differentiation in the Discussion, lines 390-403 of the revised MS; see also response to comment 10 below.

REVIEWER'S COMMENT 10: The model explains the consequence of E(z) loss. However, it would be interesting to speculate on how the dynamic E(z) PCR2 regulation enables the activation of mir-8 during homeostasis and local responses to injury. For example, the model shows how the epigenetic marks seemed up-regulated in response to DSS treatment. The authors also should try to more convincingly illustrate the proposed depletion/dilution of the epigenetic marks and its association with polyploidy or expression of the mir-8 sensor and complementarity with the epigenetic marks. Specifically, show the cells with high mir-8 sensor higher H3K27me3?

RESPONSE TO REVIEWER'S COMMENT 10: In the revised MS we have addressed the interconnected comments 7, 8 and 10 of reviewer 1 with additional experiments. Based on the available data, we also provide additional hypotheses in the revised MS as to “how the dynamic E(z) PRC2 regulation enables the activation of mir-8 during homeostasis and local responses to injury” (Discussion, lines 390-403).

In particular, we performed further immunostaining with H3K27me3 Ab in *esg>GFP* flies (Suppl. Figure 9A), and dual staining with H3K27me3 and Pros Abs in *miR-8-Sensor* flies (Suppl. Figure 13A). Please note that co-staining for H3K27me3 and DI was not possible because of incompatible fixation procedures; we thus reached conclusions about ISCs/EBs indirectly, based on the size of Pros-negative cells. The results of these experiments are summarized below:

(1) Progenitors (ISCs/EBs) have generally higher H3K27me3 fluorescence than polyploid cells (ECs); however, based on cell size we did not discern major differences in H3K27me3 fluorescence between EBs and ISCs (Suppl. Figures 9A and 13A, lines 212-215).

(2) Expression of H3K27me3 is variable in the EC population, ranging from low levels to absence of fluorescence. It appears that higher levels of H3K27me3 are encountered in aged ECs (Suppl. Fig. 9A & 9b; lines 212-215 and 217-219).

(3) miR-8-sensor fluorescence is high in ISCs and EBs (cells are, thus, miR-8 negative) but entirely absent in ECs which therefore express miR-8 (Suppl. Figure 13A, lines 250-253).

(4) Combining conclusions 1 & 3 addresses the reviewer's question: indeed, cells with high miR-8 sensor have higher H3K27me3 signal – these are the ISC/EB population. This was confirmed by H3K27me3 staining of miR-8 sensor EGFP expressing guts (Suppl. Fig. 13)

The above observations refer to the global (average) levels of H3K27me3 modification in various cell types. Please note that we do not claim that PRC2 function is inhibited globally in the EB during differentiation/tissue repair. Rather, we speculate that differentiation is triggered by **loss of methylation at specific loci, such as miR-8**, which would not be detectable by histological staining.

As requested by the reviewer, in the Discussion of the revised MS (lines 390-403) we elaborate a hypothesis involving either (a) active demethylation by dUtx or (b) displacement of PRC2 and replicational dilution following endoreplication. We speculate that immediately after ISC division, Notch signaling removes repressive histone marks placed by E(z) on early differentiation markers, like *klu* (which suppresses the mitotic cycle) and *zfh2*, through the reported Notch-mediated induction of the K27me3 demethylase dUtx. Demethylation may act in concert with damage cues (e.g. Jak/STAT and Dpp) to enable the expression of later differentiation markers, like *Sox21a*, *miR-8* and *pdm1*, leading to the transition of late EBs to ECs. During this later stage, endoreplication may further dilutes the H3K27me3 mark and consolidates the EC fate. We also speculate that the Su(H)/RBPJ transcription factor that mediates Notch signaling may gain access to Notch target gene promoters upon replicational dilution of H3K27me3 marks. Our in silico analysis of the *Drosophila* miR-8 locus indicates the

presence of four potential Su(H) binding sites (Suppl. Figure 14G), suggesting that miR-8 is a Notch target.

REVIEWER'S COMMENT 11: These interesting interactions may have implications for understanding plasticity in ageing. Do the levels of H3K27me3 change in the intestines of older flies? Other authors have shown that the defect in repair is not due to a lack of ISC mitosis but differentiation failure, which hints at a role for aberrant E(z) mediated regulation and failure to activate mir-8 in older flies...

RESPONSE TO REVIEWER'S COMMENT 11: We thank the reviewer for this insightful comment. Prompted by the reviewer's suggestion, we tested miR-8 sensor EGFP and H3K27me3 global signal in young and old guts. As shown in Suppl Figures 9A and 13B, both are upregulated in older guts. This means that the overall activity of PRC2 is increased and/or the activity of the Utx1 K27 demethylase is decreased upon aging. This is consistent with levels of miR-8 being lower (sensor is higher). The new results are discussed in lines 217-219 and 411-414 of the revised MS.

Reviewer #2

REVIEWER'S OVERALL COMMENTS: *This work studies the Polycomb components, mainly the $E(z)$, in the *Drosophila* midgut intestinal stem cell lineage. The authors used genetics, cell biology and genomics tools and they claimed that $E(z)$ is required for proper ISC differentiation. Molecularly, this regulation is through a microRNA target *mir-8*. Overall, this kind of studies would be useful for the field to understand better how chromatin regulators contribute to adult stem cell lineages. However, there are so many issues with the studies presented in this work, both technically and biologically. Technically, there is an issue of the control for RNAi (see below) and many results were shown as single images without quantifications. In addition, many loss-of-function experiments were done at one time point, which cannot give out a comprehensive picture biologically. Overall, with the current status, this work is too preliminary to be considered for publication. More rigorous and better controlled experiments are required to move forward.*

RESPONSE TO REVIEWER'S OVERALL COMMENTS: We are grateful to the reviewer for her/his positive remarks and for acknowledging the major potential contributions of our work to the field. In the revised manuscript we provide additional information and new data to address the reviewer's constructive critique with respect to controls, quantifications, and time-dependent effects of loss-of-function experiments.

REVIEWER'S COMMENT 1: *Figure 1, 2, 5, 4, 6, S1C, S2, S3, S5, according to the figure legends, "adult female flies expressing *esg-Gal4*, *tubGal80ts*, *UAS-GFP (esgts>GFP)*" which were the progenies from the cross to *w1118* strain was used as the RNAi control. This is not an appropriate control. The right control should utilize the same driver combination with a non-specific RNAi, such as a *lacZ* or *mCherry* RNAi. This way the driver drives both *UAS-GFP* and *UAS-control* RNAi or *UAS-PcG* RNAi, so that any difference of GFP is less likely due to competition for the *Gal4*. This is important for data interpretation as GFP signals have been used as a quantification method, and the GFP signals are less in the RNAi lines than in the control line.*

RESPONSE TO REVIEWER'S COMMENT 1: We thank the reviewer for giving us the opportunity to address these two important technical points.

First, in initial pilot experiments we never observed differences between *w1118* and *UAS-lacZ*. To prove our point in a more systematic manner, we have now performed detailed control experiments according to the reviewer's recommendation. In particular, we utilized the same driver combination as for *w1118* (*esg-Gal4*, *tubGal80ts*, *UAS-GFP (esgts>GFP)*) with either *lacZ* or the non-specific white gene sequence RNAi (*wRNAi*). As shown in Supplementary Figure 2, both lines gave similar results to *w1118* with respect to the percentage of GFP+ progenitors, pH3+ mitotic cells and survival upon sucrose or DSS-fed conditions. The new data is described in lines 134-137 of the revised MS and confirm the specificity of our findings.

With respect to the second technical remark, we would like to highlight the fact that the GFP measurements are qualitative, i.e. we count the GFP +ve cells and never measure the GFP

intensity which might be partly altered due to Gal4 competition. This is clarified in the Materials and Methods section of the revised manuscript, lines 455-457. These qualitative assessments fully reflect the striking differences in phenotypes between E(z)-RNAi and control intestines.

REVIEWER'S COMMENT 2. *Quite some quantifications used “representative images from posterior midguts”, what does it mean? How many representative images per midgut?*

RESPONSE TO REVIEWER'S COMMENT 2: In the Figure Legends of the revised manuscript we specify the number of images used for quantification, in line with the reviewer's recommendation.

REVIEWER'S COMMENT 3. *Fig. S3: With the sucrose, the apoptotic cells decrease significantly in E(z)-RNAi strain, any explanation for that? Still, this experiment needs to use UAS-control RNAi instead of crossing to w1118. Maybe this question can be addressed with the new control experiments.*

RESPONSE TO REVIEWER'S COMMENT 3: We thank the reviewer for bringing this point to our attention. Fig. S3 in the original submission depicted all ApopTag-positive cells in the midgut. We have now performed 3 independent repeats of the ApopTag assay focusing exclusively on GFP⁺ cells of the posterior midgut of sucrose vs DSS-fed flies. We analyzed w1118, UAS-LacZ and E(z)-RNAi young flies under the same driver combination (*esg^{ts}>GFP*), as suggested by the reviewer. The results are shown in new Supplementary Figure 5 and demonstrate absence of ApopTag-positive cells among the GFP⁺ population independently of genotype or treatment.

REVIEWER'S COMMENT 4. *All quantifications should be accompanied with supplemental tables.*

RESPONSE TO REVIEWER'S COMMENT 4: In the revised manuscript we submit a separate excel file with the quantification data.

REVIEWER'S COMMENT 5. *Scale bars need to be shown for all figure panels, currently some have but many do not.*

RESPONSE TO REVIEWER'S COMMENT 5: In the revised manuscript, we have corrected the figures to contain scale bars. We thank the reviewer for bringing this omission to our attention.

REVIEWER'S COMMENT 6. *“MARCM analysis of Su(z)12 mutated clones produced similar results (data not shown).” These data should be shown.*

RESPONSE TO REVIEWER'S COMMENT 6: In line with the reviewer's recommendation, the MARCM analysis of Su(z)12 mutated clones is shown in the Supplementary Figure 7 of the revised manuscript. This data further underscores the critical role of PRC2 in ISC differentiation.

REVIEWER'S COMMENT 7. *The MARCM results were analyzed “at 12 days after induction of recombination” therefore it is just one snapshot after inducing mitotic clones, a more careful time-course should be done to get an idea of how E(z) mutant ISCs biases differentiation toward the EC fate. Moreover, the Fig.3F showed that Delta-positive cells decline in E(z) mutant clones, image showing this should be provided as well. On the other hand, in Fig. S4, images were shown (positive with Pdm1 and negative with esg) but no quantification is provided. In summary, for the stem cell differentiation assay using different markers, images should be accompanied with quantification, and with the clonal analysis, a time-course series of data including both images and quantification should be provided to make a rigorous assay.*

An example of wt vs E(z) mutant clones is shown in Figure 3 A-E (same as in the original submission). At the reviewer's request, we have repeated the clonal analysis using more time-points (**7, 12 and 19 days** after clone induction) and the results are shown in a Supplementary Figure 6. The conclusion is that the E(z) null mutation reduces the number of DL-positive ISCs early on (at 7d). This results in smaller clones (less proliferation) at later time-points (12 and 19d), even though in the first 7d the wt and mutant clones are about the same size. We have quantified these observations in Suppl Figures 6B & 6C and describe the results in lines 169-177 of the revised MS.

With respect to the Pdm1 and Esg staining of Fig S4 in the original submission, we have revised this figure (now Suppl. Figure 8) to enumerate the percentages of these cells.

REVIEWER'S COMMENT 8. *Fig.4 shows G-TRACE results, which is very confusing. First of all, why using the same GFP for both real-time and lineage tracer. Even though the localization difference may be able to separate these two populations, based on the images it is hard to distinguish without other cellular markers. Since the nGFP signal is stronger, the cytoplasmic GFP signal became less detectable in the presence of nGFP, such as in Fig.4C-D, G-H, I-P. Why not using a difference marker, such as mCherry, for the real-time expression, as used in many other G-TRACE experiments done in different systems in Drosophila. Furthermore, the perdurance of GFP could make the real-time label beyond just “progenitor cells (ISCs and EBs)”.*

RESPONSE TO REVIEWER'S COMMENT 8. We thank the reviewer for this insightful comment. We have performed new experiments using the suggested G-TRACE line *UAS-Flp, UAS-RFP, Ubi-FRT-STOP-FRT-nEGFP* (marking red the real-time expression and green the lineage tracing). The data is shown in new Figure 4 and described in lines 188-206 of the revised MS. It confirms our original conclusion that E(z) depletion accelerates differentiation of EBs to ECs.

REVIEWER'S COMMENT 9. *“intestinal stem cells (ISCs), which are the only mitotic cells in this tissue” needs revision as recent studies showed ee cells can also divide.*

RESPONSE TO REVIEWER'S COMMENT 9: We thank the reviewer for bringing this point to our attention which refers to lines 74-76 of the original submission. The ability of pre-EEs to divide

was stated in lines 84-86 of the Introduction of the original submission. However, in line with the reviewer's recommendation and to avoid confusion, we have modified the sentence in lines 72-74 to read "*intestinal stem cells (ISCs), which are the **predominant** mitotic cells in this tissue*". In lines 81-83 we also state that: "*the EE progenitors (pre-EEs) can be generated directly by ISCs upon transient activation of the Scute (Sc) transcription factor, which typically **divide** once more before their terminal differentiation to produce a pair of EE cells*".

REVIEWER'S COMMENT 10. "miR-8 precursor (3p and 5p)", a cartoon illustrating their localization will be helpful.

RESPONSE TO REVIEWER'S COMMENT 10: In the revised MS, new (Figure 5D) we have included a cartoon illustrating the structure of miR-8-3p and 5p. The genomic localization of this miR gene is shown in Figure 5B of the initial (and present) submission.

REVIEWER'S COMMENT 11. "we observed decreased ISC mitotic rates after miR-8 overexpression, and over-accumulation of undifferentiated cells after depletion of miR-8, especially upon DSS treatment (Suppl. Figure 5B)." Where is the quantification for these statements?

RESPONSE TO REVIEWER'S COMMENT 11: In line with the reviewer's recommendation, we provide quantifications for these statements in Suppl. Figure 10 of the revised MS.

REVIEWER'S COMMENT 12. Fig.6, here other than the microRNA sensor (EGFP), is GFP signal still present from the *esg>E(z)* RNAi as shown in previous experiments and figures? Again, for both major conclusions regarding mir8 expression (Fig. 6A-B) and its functional relevance (Fig. 6C-F), only one image is shown for each without any quantifications.

RESPONSE TO REVIEWER'S COMMENT 12: We thank the reviewer for bringing these issues to our attention. Regarding the GFP signal, we made sure that in Figure 6A of the original submission (Figures 6A & 6B in revised MS), we used an *esg-Gal4* driver which does not express GFP (*esg-Gal4, tubGal80ts*), so that the only source of GFP is the *mir8*-sensor. We have added this information to the legend of Figure 6 to avoid confusion.

Detection of sensor positive epithelial cells is challenging due to extensive fluorescence of the visceral muscle surrounding the midgut, where evidently *mir-8* is not expressed (seen as bright green signal at the edges of the sections shown in A). Especially for basally localized intestinal cells (like the ISCs), which are tightly apposed to the visceral muscles, the only way to discern them is to use a marker, like DI. In the revised MS, we have included additional images of *miR-8* sensor-GFP, coupled with immunofluorescence staining with DI and Pros antibodies enabling us to visualize lower GFP expression in the ISCs of *E(z)*RNAi flies, despite the background caused by the surrounding muscles (panel 6B). Because of this extensive background, we have not

attempted to score the miR-8 sensor positive epithelial cells and base our conclusions on these qualitative data.

Regarding the quantification of the $E(z)^{731}$ -UAS-mir8sponge epistasis analysis my MARCM (Figure 6C), we have included the relevant data in a new panel in the revised MS (Figure 6D).

REVIEWER'S COMMENT 13. Fig.7, same issue that just one image is shown without any quantifications. Also, in this figure, no labeling was indicated and no scale bar was included. And only one time point was shown.

RESPONSE TO REVIEWER'S COMMENT 13: Because of space restrictions, we cannot include several pictures from each experiment. However, we would be happy to provide the raw confocal data from every experiment upon request. In the revised MS we have quantified the effects of Notch-RNAi induction on the size of $E(z)$ null mutant clones (assessed as cells per clone), in line with the reviewer's recommendation. The new data is shown in Figures 7I and 7J.

We apologize for omitting the labelling and scale bar in the confocal images of Fig. 7, we have now corrected this.

From the literature we know that Notch RNAi has a pronounced effect early, within a few days of its induction. In our $E(z)^{731}$ time course (suppl Fig. 12) we showed that the decrease in clone size is observed rather late, from 12 days onward. Therefore, for this epistasis experiment (combining the two genotypes) we selected the 10d timepoint as a compromise between an earlier timepoint (7d) and a later one (12d).

Reviewer #3

REVIEWER'S COMMENTS: *In the manuscript entitled "Polycomb-mediated silencing of miR-8 is required for maintenance of intestinal stemness in Drosophila melanogaster" authors report that the Polycomb Repressive Complex 2 (PRC2) is required for maintenance of the intestinal stem cell (ISC) pool in adult Drosophila. To elucidate the role in vivo of E(z)/PRC2 in ISC fate decisions and lineage progression they investigated its role under normal and pathological conditions. Authors clearly showed that E(z) depletion affected ISC proliferation suggesting that E(z)/PRC2 is required for the maintenance of a functional ISC pool that is critical for regenerative capacity in the Drosophila midgut. Authors showed that knocked down E(z) in intestinal stem cells and enteroblasts (ISCs/EBs) by activating RNAi expressed under the control of the conditional, temperature-sensitive *esg-Gal4*, *UAS-GFP*, *tub-GAL80ts* induces precocious differentiation into ECs. Veneti et al. showed that loss of PRC2 activity in ISCs by RNAi, results in loss of stemness and precocious differentiation of enteroblasts to enterocytes under normal and pathological conditions. Performing H3K27me3 ChIP-seq experiments authors identified direct targets of E(z)/PRC2 in ISCs/EBs. Among those targets that displayed the largest reductions in H3K27me3 upon E(z) knockdown they identified *l(2)gl*, *ph-p*, *Psc* and *miR-8* loci. Due to *mir8* involvement in the control of EB differentiation to ECs in the Drosophila intestine, authors focused their attention on *mir-8*. Of note, *miR-8* overexpression promotes enteroblast to enterocyte transition at the same extent of the E(z) knock-down. They concluded that *miR-8* is a functional target of E(z)/PRC2-mediated H3K27me3 in intestinal progenitors playing a critical role in controlling stemness versus differentiation of ISCs. Then, starting from the observation that Notch signalling is required in EBs to induce differentiation to ECs, they studied the possible link with E(z)/PRC2. Authors claimed that loss of Notch and loss of E(z) have opposite effects on stemness. Stemness defect caused by loss of E(z) function is remedied by RNAi-mediated depletion of Notch. Based on these findings the authors propose that E(z) is required for stem cell self-renewal by inhibiting spurious Notch activity. Overall, the presented work is of good quality and in some part convincing but fell short in extending the current molecular knowledge that links Notch signalling with PRC2. It has been already reported that loss of PRC2 (EED KO) affects cell plasticity at the crypt bottom by inducing differentiation towards the secretory lineage downstream of Notch activation. Thus, the link between PRC2 and the Notch signaling has been previously well documented. In my opinion no additional mechanistic insights are provided to extend this knowledge.*

RESPONSE TO REVIEWER'S COMMENTS: We thank the reviewer for the positive remarks and for acknowledging the quality of our work.

Regarding the novel aspects of our paper, we would like to re-iterate the fact that the *in vivo* actions of E(z)/EZH2 in ISC fate decisions and lineage progression have not yet been elucidated. As indicated in the Introduction of the original (and present) submission, lines 98-99, whereas EED ablation in the **entire** intestinal epithelial compartment in the mouse has indeed been shown to affect cell plasticity at the bottom of the crypt, the role of EED **specifically in the ISCs** remains unknown. Most importantly, the effect of EED observed in the

mouse intestine (increase in hormone-secreting cells) was different than what we observed by targeting E(z) in Drosophila ISCs in this study (loss of stem cells). Indeed, such pleiotropically deployed factors need to be studied in many tissues (let alone many species) in order to get a complete picture of their biological activities.

Along the same lines, with regard to the PRC2-Notch interplay, we believe that the demonstration of a relationship between two highly pleiotropic pathways in one context does not predetermine their relation in a different context. For example, although Notch activation in the mouse intestine seems to use PRCs to antagonize a secretory lineage (via Hes1 and PRC2-mediated repression of Atoh1), this is not the case in the Drosophila intestine where PRC2 is somehow needed to promote (not to suppress) the enteroendocrine (EE) fate (PMID: 33724181).

Even in the same species and organ, PRC2 and Notch can each have a multitude of roles in different cells at different times. In the Drosophila intestine, it is not known whether the well-described activity of Notch has any connection with PRC2-mediated chromatin remodeling. Although the main focus of our manuscript is not the PRC2-Notch interplay, we have provided some data of a possible connection in the experiments of Fig. 7 and Suppl. Fig. 7.

In the revised MS we provide **additional evidence for a functional interaction between E(z) and Notch** signaling in intestinal progenitor cells by showing that the known reduction in survival of flies caused by Notch depletion in ISCs/EBs is partly alleviated upon E(z) knock-down (Figure 7K in revised MS). This result adds weight to our conclusion that E(z) and Notch are somehow interconnected in the regulation of intestinal stem cell self-renewal in Drosophila and is a topic worth studying more thoroughly in the future. We hope that these explanatory comments and new data address the reviewer's concerns.

REVIEWER COMMENTS

Reviewer #1 (Remarks to the Author):

My comments/concerns have all been addressed in the revisions.

Given that the authors have performed their studies only in mated females, I strongly encourage the authors to follow the 'Sex and Gender Equity in Research – SAGER – guidelines' and to include sex in the title of the manuscript and/or include this relevant information in the abstract.

Reviewer #2 (Remarks to the Author):

The revision has added a few more experiments, such as using a new marker for G-TRACE, but several outstanding concerns raised previously have not been addressed. Therefore, I do not think the revision reaches the rigor I would like to see. Here are just a few examples:

1. The authors did some revision with additional experiments. However, I do not think they have sufficiently addressed the concerns raised previously. For example, for the proper control experiments throughout the manuscript, the authors just added UAS-wRNAi and UAS-lacZ with *esg-Gal4* and compared them with *w1118 + eag-Gal4*, but this is NOT the issues I raised concerning their experimental design, these controls need to be done when they use the *esg-Gal4*, *tubGal80ts*, UAS-GFP with the different UAS-E(z) RNAi or UAS-Su(z)12 RNAi, etc., since the competition for the Gal4 driver is applicable when more than one UAS-transgenes or RNAi are present with one Gal4, but not just at the background with one UAS-wRNAi or UAS-lacZ. In another word, I do not think these additional experiments are the appropriate controls to address the concerns raised previously and most of the data presented in this work still lack the proper control. For example, in Figure 2 legend "Flies with *esgts>GFP* were crossed to either UAS-E(z)-RNAi to knockdown E(z) in intestinal progenitor cells or to *w1118* to generate the respective control genotype (control)." Here the control should not be the cross to *w1118* but to UAS-lacZ or UAS-wRNAi. For some other figures, how the control experiments were performed was not explained sufficiently, such as Figure 4.
2. Regarding this explanation in the rebuttal letter "With respect to the second technical remark, we would like to highlight the fact that the GFP measurements are qualitative, i.e. we count the GFP +ve cells and never measure the GFP intensity which might be partly altered due to Gal4 competition. This is clarified in the Materials and Methods section of the revised manuscript, lines 455-457." What does the "GFP +ve cells" stand for? I cannot find proper explanation of this in lines 455-457.
3. The DI immunostaining is not always clear, the DI-nLacZ has a nuclear signal and should be used to confirm the DI staining results, such as those shown in Fig. 2.

Reviewer #3 (Remarks to the Author):

Upon careful consideration of the revisions made by the authors, I am pleased to note that the manuscript has undergone significant improvement. The authors have diligently addressed the concerns raised during the initial review, and the subsequent modifications have notably enhanced the overall quality and clarity of the manuscript.

The commendable efforts invested in refining the content have positively contributed to the manuscript's coherence and scientific rigor. The revisions reflect a dedicated response to the feedback provided, and it is evident that the authors have strived to elevate the manuscript to meet the high standards of Nature Communications.

In light of the substantial improvements observed, I am inclined to recommend acceptance of the manuscript for publication. However, I acknowledge the importance of a comprehensive and unbiased evaluation process. Should my fellow reviewers concur with my assessment, I would be amenable to endorsing the acceptance of the manuscript for publication in Nature Communications.

Manuscript (MS) NCOMMS-23-08936 REVISED

Veneti et al, *Polycomb-mediated silencing of miR-8 is required for maintenance of intestinal stemness in Drosophila melanogaster*

Response to Reviewers' comments

REVIEWER 1

Reviewer's comment: *My comments/concerns have all been addressed in the revisions. Given that the authors have performed their studies only in mated females, I strongly encourage the authors to follow the 'Sex and Gender Equity in Research – SAGER – guidelines' and to include sex in the title of the manuscript and/or include this relevant information in the abstract.*

RESPONSE: We thank the reviewer for the positive feedback and for acknowledging the thorough revisions made in response to his/her comments. In line with the reviewer's recommendation and the SAGER guidelines, we have included the sex-related information in the Abstract of the revised manuscript, line 30.

REVIEWER 2

Reviewer's comment 1: *The revision has added a few more experiments, such as using a new marker for G-TRACE, but several outstanding concerns raised previously have not been addressed. Therefore, I do not think the revision reaches the rigor I would like to see. Here are just a few examples:*

The authors did some revision with additional experiments. However, I do not think they have sufficiently addressed the concerns raised previously. For example, for the proper control experiments throughout the manuscript, the authors just added UAS-wRNAi and UAS-lacZ with esg-Gal4 and compared them with w1118 + eag-Gal4, but this is NOT the issues I raised concerning their experimental design, these controls need to be done when they use the esg-Gal4, tubGal80ts, UAS-GFP with the different UAS-E(z) RNAi or UAS-Su(z)12 RNAi, etc., since the competition for the Gal4 driver is applicable when more than one UAS-transgenes or RNAi are present with one Gal4, but not just at the background with one UAS-wRNAi or UAS-lacZ. In another word, I do not think these additional experiments are the appropriate controls to address the concerns raised previously and most of the data presented in this work still lack the proper control. For example, in Figure 2 legend "Flies with esgts>GFP were crossed to either UAS-E(z)-RNAi to knockdown E(z) in intestinal progenitor cells or to w1118 to generate the respective control genotype (control)." Here the control should not be the cross to w1118 but to UAS-lacZ or UAS-wRNAi. For some other figures, how the control experiments were performed was not explained sufficiently, such as Figure 4.

RESPONSE: We would like to thank the reviewer for bringing this issue to our attention. In the previous revision of our paper we provided exactly the requested control experiments and shown

the results in Supplementary Figure 2. **We apologize that these modifications were not conveyed with sufficient clarity with respect to the genotypes generated and evaluated.**

We understand that the reviewer is concerned about the possible "dilution" of Gal4 molecules when multiple UAS targets are present. For this reason, we crossed *esg-Gal4 Tub-Gal80ts UAS-GFP* (one UAS) with either *w¹¹¹⁸* (zero UAS), *UAS-lacZ* (+ one more UAS) or *UAS-wRNAi* (+ one more UAS). As shown in Supplementary Figure 2, the combinations comprising two UAS (*UAS-lacZ+UAS-GFP* and *UAS-wRNAi+UAS-GFP*) were **no different** in terms of gut histology, phospho-Histone 3 positive (pH3⁺) cells or survival than the one UAS (*UAS-GFP* only) upon sucrose or DSS-fed conditions. Please note that the experiment in Fig. 2B (and 1C,D) used the exact same driver (*esg-Gal4 Tub-Gal80ts UAS-GFP*) crossed to *UAS-E(z)-RNAi* (i.e. two UAS: *GFP + E(z)RNAi*).

In this revision we provide additional data to further exclude the possibility that "dilution" of Gal4 molecules impact the observed phenotypes. Specifically, we assessed the percentage of DI-positive cells in *UAS-wRNAi+UAS-GFP* (two UAS) and compared it to *UAS-GFP w¹¹¹⁸* (one UAS). As shown in the new Supplementary Figure 4D,E *versus* the original Figure 2C,D, the percentage of DI-positive cells is not affected by the number of UAS.

Since the addition of the second UAS in the experiments of Supplementary Figure 2B,C and 4D,E did not affect the results when compared to Figures 2C,D and Supplementary 2A, we did not feel compelled to repeat all other control experiments with the additional UAS transgene (*w-RNAi* or *lacZ*). For example, the controls in Figure 4 used an *esg^{ts} Tub-Gal80ts* driver (no *UAS-GFP*) crossed to the *G-trace* stock (*UAS-FLP UAS-dsRed ubi>stop>GFP*) in the presence of *UAS-E(z)-RNAi* (three UAS transgenes total) or in its absence (two UAS-transgenes).

We apologize for the insufficient detail given in the legends of Figure 4 and Supplementary Figure 2 regarding these genotypes and have corrected it (lines 595-598 and lines 686-693).

REVIEWER'S COMMENT 2. *Regarding this explanation in the rebuttal letter "With respect to the second technical remark, we would like to highlight the fact that the GFP measurements are qualitative, i.e. we count the GFP +ve cells and never measure the GFP intensity which might be partly altered due to Gal4 competition. This is clarified in the Materials and Methods section of the revised manuscript, lines 455-457." What does the "GFP +ve cells" stand for? I cannot find proper explanation of this in lines 455-457.*

RESPONSE: We apologize for wrongly stating the revised lines in the text: the correct line numbers where the revised information was provided were 445-447 (457-459 in this revision). There, we state that: "Progenitor cells were quantified by counting GFP-positive cells. GFP was assessed in a qualitative manner, no attempt was made to measure GFP intensity which might be altered due to competition for the Gal4 in RNAi lines." In the rebuttal letter, we had left the "+ve" shorthand (for the word "positive") by oversight. We hope that this clarifies the issue. [The above quote was our response to the original concern of the reviewer that stated "*This is important for data interpretation as GFP signals have been used as a quantification method, and the GFP signals are less in the RNAi lines than in the control line.*". In short, we did not use the level of GFP signal as a quantification method, we simply counted GFP-positive cells].

REVIEWER'S COMMENT 3. *The DI immunostaining is not always clear, the DI-nLacZ has a nuclear signal and should be used to confirm the DI staining results, such as those shown in Fig. 2.*

Regarding Figure 2 of the original manuscript, the levels of DI are weak (which is the case in the literature from many other labs, see for example PMID: 30837466 – clearly Delta is expressed at low levels), yet above background and reproducible.

We would like to note that the methodology we employed to assess the **endogenous DI** protein by using a specific antibody and not a lacZ enhancer trap line, is the approach of choice and is widely utilized in the majority of pertinent published papers, including two instrumental studies about Delta-Notch signaling in ISCs, published in *Science* (PMID: 17303754 and PMID: 26586765). The Delta antibody has also been used to confirm the ISC specificity of the DI-lacZ and DI-Gal4 lines (PMID: 20681020).

For these reasons, and the fact that the fly strain DI-nLacZ recommended by the reviewer is not available through a public repository, **we opted for repeating the DI staining**, as follows. We crossed *esg-Gal4 Tub-Gal80ts UAS-GFP* with either *UAS-wRNAi* or *UAS-E(z)RNAi* and at 7 days post induction, we immunostained posterior midguts exactly as described in Figure 2 with the exception of using only the DI antibody (i.e. rather than performing dual DI and Pros staining). The results of this analysis are presented in the new Supplementary Figure 4D,E, reported in lines 156-157, and unequivocally **confirm the conclusions of the original Figure 2**.

This ancillary information comes to verify what we have rigorously shown in many experiments already and by using different genetic stem cell tracking tools, namely that PRC2 downregulation gradually eliminates stem cells (Figures 1, 2 & 3; Suppl. Fig. 1C, 3 & 4).

We hope that the addition of the new data and our explanatory comments fully address the reviewer's critique.

REVIEWER 3

Upon careful consideration of the revisions made by the authors, I am pleased to note that the manuscript has undergone significant improvement. The authors have diligently addressed the concerns raised during the initial review, and the subsequent modifications have notably enhanced the overall quality and clarity of the manuscript.

The commendable efforts invested in refining the content have positively contributed to the manuscript's coherence and scientific rigor. The revisions reflect a dedicated response to the feedback provided, and it is evident that the authors have strived to elevate the manuscript to meet the high standards of Nature Communications.

In light of the substantial improvements observed, I am inclined to recommend acceptance of the manuscript for publication. However, I acknowledge the importance of a comprehensive and unbiased evaluation process. Should my fellow reviewers concur with my assessment, I would be amenable to endorsing the acceptance of the manuscript for publication in Nature Communications.

RESPONSE: We are delighted for the positive feedback from Reviewer #3, acknowledging that our comprehensive revisions have significantly contributed to the manuscript's coherence and scientific rigor.

REVIEWERS' COMMENTS

Reviewer #2 (Remarks to the Author):

The authors have provided more clarifications on the issues raised previously. A few changes are still needed before acceptance:

1. For the proper controls, the authors should provide a direct comparison using zero UAS, one UAS, two UASs, on the same figure with statistics analyses. Currently, these data are scattered throughout the manuscript in different figure panels. Additionally, the data points for Supple. figure 4E are rather scarce, need to include more and place all controls together in one panel for direct comparison.
2. Again, I cannot find these statements in "(457-459 in this revision)", as the author indicated in the Response to Reviewers' comments. Instead, this statement seems to be at 447-450. Please confirm and it is also very annoying to have this simple information wrong twice.

Response to Reviewer's comments

Reviewer #2 (Remarks to the Author):

REVIEWER'S COMMENT: *The authors have provided more clarifications on the issues raised previously. A few changes are still needed before acceptance:*

1. For the proper controls, the authors should provide a direct comparison using zero UAS, one UAS, two UASs, on the same figure with statistics analyses. Currently, these data are scattered throughout the manuscript in different figure panels. Additionally, the data points for Supple. figure 4E are rather scarce, need to include more and place all controls together in one panel for direct comparison.

RESPONSE TO REVIEWER'S COMMENT: In the revised MS we have included a direct comparison of different UAS in **modified Supplementary Figure 2** with statistical analysis, as requested by the reviewer. Additionally, we have modified **Supplementary Figure 4E** to include more measurements as well as all controls in one panel, as requested by the reviewer.

REVIEWER'S COMMENT 2. *Again, I cannot find these statements in "(457-459 in this revision)", as the author indicated in the Response to Reviewers' comments. Instead, this statement seems to be at 447-450. Please confirm and it is also very annoying to have this simple information wrong twice.*

RESPONSE TO REVIEWER'S COMMENT: We apologize for the mistake. We confirm that this statement was in lines 447-450 of the previous revision.